# Identification of Average Outcome under Interventions in Confounded Additive Noise Models

**Muhammad Qasim Elahi**                                  *elahi0@purdue.edu*
*School of Electrical and Computer Engineering*
*Purdue University*

**Mahsa Ghasemi**                                        *mahsa@purdue.edu*
*School of Electrical and Computer Engineering*
*Purdue University*

**Murat Kocaoglu**                                       *mkocaoglu@jhu.edu*
*Department of Computer Science*
*Johns Hopkins University*

**Reviewed on OpenReview:** *https://openreview.net/forum?id=y5YnHzLf1d*

## Abstract

Additive noise models (ANMs) are an important framework studied in causal inference. Most existing works assume causal sufficiency, i.e., no latent confounders, and focus on observational data for inference and discovery. In this paper, we study confounded ANMs, where treatment and outcome variables are influenced by an unobserved confounder, and we consider interventions on treatment variables to identify various causal effects. We introduce a novel approach for estimating the average outcome under interventions (AOIs) for any subset of treatment variables and show that a small set of interventional distributions is sufficient to estimate all AOIs. Furthermore, we propose a randomized algorithm that reduces the number of required interventions to poly-logarithmic in the number of nodes. We also demonstrate that these interventions suffice to recover the causal structure among the observed variables. Consequently, a poly-logarithmic number of interventions is sufficient to infer the causal effects of any subset of treatments on the outcome in confounded ANMs with high probability, even when the causal relationships among treatments are unknown. Simulation results indicate that our method can accurately estimate all AOIs in finite-sample settings, and we further demonstrate its practical significance using semi-synthetic data.

## 1 Introduction

Discovering cause and effect relationships is a paramount objective in data sciences, artificial intelligence, and machine learning. Through meticulous scientific study, identifying causal relations enhances the validity and generalizability of corresponding effects across various conditions (Lee & Bareinboim, 2020). causal relations are deemed desirable and valuable for constructing explanations and for contemplating novel interventions that were never experienced before (Pearl, 2009; Lee & Bareinboim, 2020). A significant area of interest in causal reasoning is the use of observations and a series of interventions to identify various causal effects in a structural causal model (SCM) (Pearl, 2009; Peters et al., 2017; Kandasamy et al., 2019; Stovitz & Shrier, 2019). Based on the framework developed by Pearl, an SCM consists of endogenous and exogenous variables and functional mappings that determine the values of endogenous values (Pearl, 2009). Every SCM can also be represented as a directed acyclic graph (DAG), with variables represented as nodes, and relationships between variables represented as edges. Many fields, including medicine, advertising, economics, and others, utilize multiple interventions (Worrall, 2011; Rizzi & Pedersen, 1992). To account for unobservable variables, interventional data and causal inference techniques may be required (Tian & Pearl, 2002). While randomized

controlled experiments can be used to identify causal relationships, they are often costly or unfeasible (Farmer et al., 2018). Estimating treatment effects from observational data has gained attention due to limitations and ethical concerns of randomized experiments (Yao et al., 2021). Observational data enables researchers to investigate causal effects without actually running the experiments (Yao et al., 2021). If there are no latent variables, it is possible to identify all causal effects from the observational distribution (Greenland & Robins, 1986; Spirtes et al., 2000). When there are unobserved variables, the observational distribution may not be enough to identify causal effects due to potential confounding.

In additive noise models (ANMs), it is assumed that the noise in the structural equations of variables is additive. If the noise terms in the ANM are statistically independent for different variables, it implies that there is no confounding present (Peters et al., 2014). However, when the noise terms are not independent and have a joint distribution, the ANM is considered confounded (Jeunen et al., 2022). Estimating causal effects from purely observational data in the presence of unobserved confounders, which may affect both the treatment and the outcome, is a challenging task. Independence of noise terms is a sufficient condition for no confounding under the assumption of a correctly specified additive noise model. Dependence among noise terms may arise either from hidden confounding or from model misspecification, such as assuming linearity when the true data-generating process is nonlinear. In this work, we focus solely on correctly specified ANMs and do not address model misspecification.

In this work, we aim to identify all AOIs in confounded ANMs with multivariate Gaussian noise. Identifying the causal effect of any subset of treatments is crucial because real-world interventions often occur in combinations, and their joint effects may differ from individual effects. This is particularly relevant in medicine, public policy, education, agriculture, and marketing, where multiple interventions are applied simultaneously, and the goal is to identify the best or optimal interventions through a series of experiments. The ability to determine effects for all possible intervention combinations, i.e., all possible subsets of treatments, using a much smaller number of interventional data sets, can be highly useful in such scenarios. Estimating these effects enables better decision-making, resource allocation, and the design of effective intervention strategies. To achieve this, we make specific assumptions about the noise in our analysis. We assume that the noise follows a multivariate Gaussian distribution. Furthermore, we assume that a treatment variable cannot be a descendant of the outcome variable. A probability distribution is said to be faithful to a graph if and only if every conditional independence statement in the distribution can be inferred from the graph via *d*-separation, a graphical criterion that determines whether two sets of variables are conditionally independent given a third set. Faithfulness is a commonly used assumption in existing work on causal discovery Kocaoglu et al. (2017); Hauser & Bühlmann (2014). We assume that the faithfulness assumption holds for both the observational and post-interventional distributions.

- We propose an algorithm for estimating AOIs of any subset of treatments in confounded ANMs from a small number of interventional data sets. We identify conditions that the set of available interventional data sets needs to satisfy to enable this.

- We propose a randomized algorithm that can be used to identify poly-logarithmic number of interventions (in the number of variables) that satisfy our identifiability condition for estimating all AOIs in confounded ANMs. We also show that the same set of interventions are sufficient for discovering the causal graph if it is not available. Specifically, the algorithm requires $\mathcal{O}(8d_{max} \log^3 n)$ interventions, where $d_{max}$ is the largest degree in the causal graph, and $n$ is number of treatment variables in the graph.

- Using randomly generated causal graphs, we demonstrate the effectiveness of our approach in accurately inferring AOIs, even when working with limited data samples from experiments. We showcase the practical significance of our inference scheme by evaluating it on semi-synthetic data as well.

## 2 Related Work

In various disciplines, such as medicine, business, and advertising, the ability to provide causal explanations and answer causal queries is crucial for making informed decisions, understanding relationships between

variables, and evaluating the effectiveness of an intervention (Nabi et al., 2022; Álvarez-Martínez & Pérez-Campos, 2004). This growing body of research highlights the increasing importance of causal inference across various fields. Causal inference is an important tool to infer the effect of interventions without actually carrying them out. An important area of research is to determine the conditions under which certain causal effects can be estimated from the observational data and causal information transferred from other experiments. For example, Bareinboim & Pearl (2012) provides an algorithm for fusing observational and experimental data to estimate a causal query. In Hoyer et al. (2008), authors show that as long as the noise is additive, the linear non-Gaussian causal discovery framework can be extended to nonlinear functional dependencies among the variables.

Imposing certain limitations on the underlying causal structure can render causal quantities identifiable, even when they are generally unidentifiable. The model being an ANM is a common practical assumption. The ANM assumes that the latent variable is additive in the structural equation of the observed variables (Maclaren & Nicholson, 2019; Kap et al., 2021). ANMs do allow non-linear mappings between the observed variables. In Peters et al. (2014), the authors demonstrate that under mild conditions, the causal graph of a model with an additive noise structure can be identified from the joint observational distribution. They assume that the additive noise follows a joint normal distribution and is independent. Additionally, the structural equations in the model are nonlinear and atleast thrice differentiable (Peters et al., 2014). The paper Rolland et al. (2022) proposes a method for recovering causal graphs in a non-linear additive (Gaussian) noise model. The proposed approach utilizes score-matching algorithms as a building block to design a new generation of scalable causal discovery methods.

The work by Zhang et al. (2020) proposes a simultaneous approach to discovery and identification for linear models. The proposed symbiotic approach leverages information gained from causal discovery to aid inference, and vice versa. The paper Saengkyongam & Silva (2020) investigates the identifiability of joint intervention from the conjunction of observational and single-variable interventional data. The authors show that without any restriction on the structure of the SCM, it is not identifiable. The complementary question is addressed by Jeunen et al. (2022), which is to estimate the causal effect of a single treatment from a conjunction of observational and joint intervention data regimes under the assumption that treatments do not have causal effects on one another, and Gaussian noise has a zero mean. They show that the observational distribution and joint interventional distribution enable the identification of all AOIs in this scenario. Our work is focused on more general ANMs without any additional restrictions on the structural equations, where treatments can have causal effects on one another, and the mean of Gaussian noise is non-zero.

The main contribution of the paper is to identify the set of interventional distributions sufficient to determine all possible AOIs for all possible subsets of treatment variables in confounded ANMs. Our main identification scheme requires knowledge of the underlying causal graph for the ANM. In many practical scenarios, the causal graph is not available. To address this situation, we modify one of the algorithms proposed in Kocaoglu et al. (2017) to not only learn the observed graph structure in the ANM between the treatments and the outcome but also simultaneously return the interventional data sets for the inference task. Although there are many other proposed approaches to learning the causal graph with latent variables, including but not limited to Addanki et al. (2020); Akbari et al. (2022), our current choice is due to the fact that we show we can use the same algorithm to simultaneously learn the causal graph and return the interventional data sets that satisfy a criteria enabling identification for all possible AOIs. This allows us to propose a randomized algorithm that can be used to identify a poly-logarithmic number of interventions satisfying our identifiability condition for estimating all AOIs in confounded ANMs, even when the underlying causal graph is unknown.

## 3 Background

A Structural Causal Model (SCM) can be defined as the tuple $\mathcal{M} = \langle \{\mathbf{X}, Y\}, \mathbf{U}, \mathcal{F}, P_{\mathbf{u}} \rangle$, where $\{\mathbf{X}, Y\}$ are the observed/endogenous variables separated into treatment set $\mathbf{X}$ and outcome $Y$, $\mathbf{U}$ are the exogenous variables (possibly confounders) with $P_{\mathbf{u}}$ defining their joint probability distribution and $\mathcal{F}$ are the structural equations (Pearl, 2009). The SCM can be modeled as a DAG $G = (\mathbf{V}, \mathbf{E})$, where vertices $\mathbf{V}$ correspond to observed variables and edges $\mathbf{E}$ correspond to the causal relationships. In the DAG a directed edge $(X \rightarrow Y)$ denotes that $X$ is a direct cause of $Y$. On the other hand, a bi-directed edge $(X \leftrightarrow Y)$ indicates the presence

of an unobserved confounder as the common parent of $X$ and $Y$. The set of nodes having outgoing edges to a particular node $X_i$ is termed as parents of that node, denoted by $Pa(X_i)$. All the endogenous variables including the $X_i$ and outcome $Y$ can be written as a function of the parent variables and the corresponding latent variable. Without loss of generality we assume that all treatments $X_i$ are parents of the outcome $Y$ and the noise variables are correlated.

$$X_i := f_i(Pa(X_i), U_i) \quad , \quad Y := f_Y(\mathbf{X}, U_Y) \tag{1}$$

We focus on an important class of SCMs called confounded ANMs where the latent variables appear as additive noise in the structural equations. Similar to Jeunen et al. (2022), we assume that the latent variables, jointly represented as $\mathbf{U}$, follow a multivariate Gaussian distribution.

$$X_i := f_i(Pa(X_i)) + U_i \quad , \quad Y := f_Y(\mathbf{X}) + U_Y \tag{2}$$

For an SCM $\mathcal{M}$, an intervention $do(\mathbf{W})$ on a set of variables $\mathbf{W} \subseteq \mathbf{X}$ refers to fixing their values irrespective of the values that the parent variables $Pa(\mathbf{W})$ take. This can be encoded by replacing the original structural equations of $\mathbf{W}$ with some constant, which induces a submodel $\mathcal{M}_{do(\mathbf{W})}$. The intervention on a set of variables $\mathbf{W}$ is equivalent to breaking all the incoming edges into $\mathbf{W}$ in the corresponding causal graph $G$. We denote the modified causal graph with incoming edges to set of nodes $\mathbf{W}$ removed as $G_{\overline{\mathbf{W}}}$. For a given intervention $do(\mathbf{W})$, the distribution observed over the rest of the variables $P[\mathbf{X} \setminus \mathbf{W} | do(\mathbf{W})]$ is called the interventional distribution. We denote an interventional distribution in an SCM $\mathcal{M}$ with a set of endogenous variables $\mathbf{X}$ as $P^{\mathcal{M}}[\mathbf{X} \setminus \mathbf{W} | do(\mathbf{W})]$. In our work, we define the average outcome under intervention (AOI) as $\mathbb{E}[Y \mid do(\mathbf{W} = \mathbf{w})]$, where $\mathbf{W}$ is any subset of the treatment variables, i.e., $\mathbf{W} \subseteq \mathbf{X}$. In previous literature, the average treatment effect (ATE), also sometimes referred to as the average causal effect (ACE), is defined as the difference: $\mathbb{E}[Y \mid do(\mathbf{W}_1 = \mathbf{w}_1)] - \mathbb{E}[Y \mid do(\mathbf{W}_2 = \mathbf{w}_2)]$, where $\mathbf{W}_1, \mathbf{W}_2 \subseteq \mathbf{X}$ (Varian, 2016; Naimi et al., 2017; Frauen et al., 2023). Since we propose a machinery to identify the AOIs of the form $\mathbb{E}[Y \mid do(\mathbf{W} = \mathbf{w})]$, our proposed machinery can also be used to identify the classic ATE or ACE, which is defined as the difference between two AOIs.

## 4 Identifiability of Causal Effects for Additive Noise Models

An estimand is identifiable if its value can be uniquely determined from unlimited data samples. This means that any two models that agree on data must also agree on the estimand when the query is identifiable. Our goal is to construct a set of intervention targets denoted by $\mathcal{I}$, such that we can identify any AOI of the form $\mathbb{E}[Y | do(\mathbf{W})]$ where $\mathbf{W} \subseteq \mathbf{X}$.

**Definition 1** *Given a causal graph $G$, and a collection of interventional distributions $\{P(\boldsymbol{X} | do(\boldsymbol{W}_i))\}$, an interventional distribution $P(y | do(\boldsymbol{z}))$ is not identifiable iff among all models that entail the causal graph $G$ and the interventional distributions $\{p(\boldsymbol{X} | do(\boldsymbol{W}_i))\}$, there exists two models $\mathcal{M}_1, \mathcal{M}_2$ where $P^{\mathcal{M}_1}[y | do(\boldsymbol{z})] \neq P^{\mathcal{M}_2}[y | do(\boldsymbol{z})]$.*

The paper Jeunen et al. (2022) shows that for the confounded ANMs where there is no direct causal relationship among the treatments, all AOIs can be identified from the observational distrbution and joint interventional distribution on all treatments. However, in a general cases where treatments can have direct causal effects on other variables, the previously mentioned result may not be applicable. We can show this with help of a simple example. Consider a intervention target set $\mathcal{I}$ that includes two interventions: the empty intervention $\emptyset$ for observational data and the joint intervention $\mathbf{X}$ for simultaneous interventions on all variables. We show that this set $\mathcal{I} = \{\emptyset, \mathbf{X}\}$ is not sufficient to identify all the AOIs in ANMs.

The two SCMs $\mathcal{M}_1, \mathcal{M}_2$ in Table 1 agree on the interventional distributions for targets in the set $\mathcal{I} = \{\emptyset, \{X_1, X_2\}\}$, but they can disagree on other interventional distributions. The interventional distribution $do(X_1 = x_1, X_2 = x_2)$ is the same for both $\mathcal{M}_1$ and $\mathcal{M}_2$, i.e. $\mathcal{N}(x_1 x_2, 1)$. Similarly, the observational distribution for $X_1$ is the same, i.e. $X_1 \sim \mathcal{N}(0, 1)$. For both models, the variable $X_2$ is a zero mean Gaussian with a variance of 3, i.e. $X_2 \sim \mathcal{N}(0, 3)$. Also, the covariance between $X_1$ and $X_2$ is the same across both models, i.e. $Cov(X_1, X_2) = 1.5$. Consequently, the observational distribution is the same for

Table 1: Two SCMs where $(U_1, U_2, U_Y)_{\mathcal{M}_1} \sim \mathcal{N}(0, \Sigma_{\mathcal{M}_1})$ and $(U_1, U_2, U_Y)_{\mathcal{M}_2} \sim \mathcal{N}(0, \Sigma_{\mathcal{M}_2})$.

| $\mathcal{M}_1$ | $\mathcal{M}_2$ |
|---|---|
| $X_1 = U_1$ | $X_1 = U_1$ |
| $X_2 = X_1 + U_2$ | $X_2 = 1.2X_1 + 1.2U_2$ |
| $Y = X_1 X_2 + U_Y$ | $Y = X_1 X_2 + U_Y$ |

$$\Sigma_{\mathcal{M}_1} = \begin{bmatrix} 1 & \frac{1}{2} & 0 \\ \frac{1}{2} & 1 & 0 \\ 0 & 0 & 1 \end{bmatrix} \Sigma_{\mathcal{M}_2} = \begin{bmatrix} 1 & \frac{1}{4} & 0 \\ \frac{1}{4} & \frac{7}{12} & 0 \\ 0 & 0 & 1 \end{bmatrix}$$

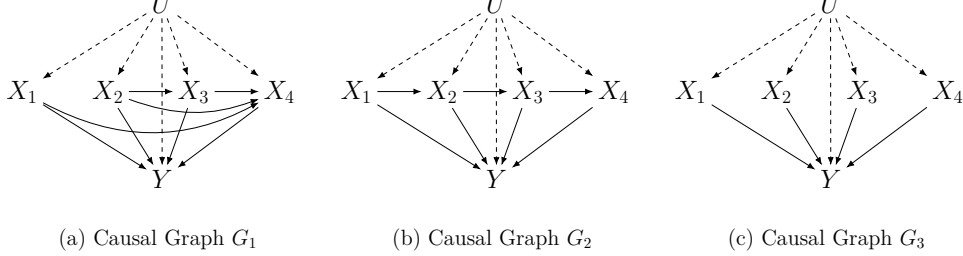

(a) Causal Graph $G_1$      (b) Causal Graph $G_2$      (c) Causal Graph $G_3$

Figure 1: Causal graphs for different confounded additive noise models.

both $\mathcal{M}_1$ and $\mathcal{M}_2$. Therefore, both SCMs agree on the interventional distributions for targets in the set $\mathcal{I}$ defined earlier. However, we have $P^{\mathcal{M}_1}[X_2|do(X_1)] \neq P^{\mathcal{M}_2}[X_2|do(X_1)]$ which implies that interventional distribution $do(X_1)$ is unidentifiable from interventional distributions in $\mathcal{I}$. This implies that we need a bigger intervention target set $\mathcal{I}$ to make all average causal identifiable for confounded ANMs. This example demonstrates that it is possible to create two distinct SCMs that agree on a given observational distribution and intervention on all treatments $do(\mathbf{X})$, but have differences in either their structural equations or noise parameters. Consequently, these models will disagree on some interventional distributions. We show that with additional interventional datasets, including interventions on the parent sets of the treatment variables, we can make all possible AOIs in confounded ANMs identifiable.

## 4.1 Core Interventions for Identification

We first prove a sufficient condition to make all AOIs identifiable for an ANM with multivariate Gaussian noise in Theorem 1. Later, we extend this result to provide the necessary and sufficient set of conditions that interventional data sets need to satisfy to ensure the identifiability of all AOIs in confounded ANMs with Gaussian noise in Theorem 2.

**Theorem 1 (Identifiability of AOIs in confounded ANMs)** *Let $\langle \boldsymbol{V} = \{\boldsymbol{X}, Y\}, \boldsymbol{U}, \mathcal{F}, P_{\boldsymbol{u}} \rangle$ be an SCM where $X_i = f_i(Pa(X_i)) + U_i$, $Y = f_Y(\boldsymbol{X}) + U_Y$ and $P_{\boldsymbol{u}} \sim \mathcal{N}(\mu, \Sigma)$. All estimands of the form $\mathbb{E}[Y|do(\boldsymbol{W})]$ where $\boldsymbol{W} \subseteq \boldsymbol{X}$ are identifiable from the conjunction of the two data regimes; the observational distribution and interventional distributions given that the collection of interventional distributions includes $P[Y|do(\boldsymbol{X})]$ and $\{P[\boldsymbol{V} \setminus Pa(X_i)|do(Pa(X_i))], \forall i = 1....n\}$.*

A sketch of the proof of Theorem 1 is provided here, with a detailed proof available in the supplementary material. Consider two sets of treatment variables such that: $\mathbf{X}_{int}, \mathbf{X}_O \subseteq \mathbf{X}$ and $\mathbf{X}_O = \mathbf{X} \setminus \mathbf{X}_{int}$. The structural equation $f_Y(\mathbf{X})$ is identified from the interventional distribution $do(\mathbf{X})$. Consequently, the conditional AOI of the form $\mathbb{E}[Y|do(\mathbf{X}_{int}), \mathbf{X}_O]$ can be identified as follows:

$$\mathbb{E}[Y|do(\mathbf{X})] = f_Y(\mathbf{X}) + \mathbb{E}[U_y]$$

$$\mathbb{E}[Y|do(\mathbf{X}_{int}), \mathbf{X}_O] = f_Y(\mathbf{X}_{int}, \mathbf{X}_O) + \mathbb{E}[U_y|\mathbf{X}_O, do(\mathbf{X}_{int})]$$

$$\mathbb{E}[U_y|\mathbf{X}_O, do(\mathbf{X}_{int})] = \mathbb{E}[U_y|\mathbf{U}_O = \mathbf{X}_O - \mathbf{f}_O(Pa(\mathbf{X}_O))]$$

We can estimate the conditional expectation $\mathbb{E}[U_y \mid \mathbf{X}_O, do(\mathbf{X}\text{int})]$ by identifying the structural equations for $\mathbf{X}_O$ and the noise covariance matrix. To do so, we utilize interventional distributions of the form

$do(Pa(\mathbf{X}_O))$ and rearrange equation 2 to estimate the noise covariance matrix. The detailed proof for this inference mechanism is provided in the supplementary material. The intervention target set $\mathcal{I}$ for identifying all possible AOIs in ANMs includes both the observational and interventional distributions: $do(\mathbf{X})$ and $do(Pa(X_i))$. Our theoretical results assume that the latent confounder follows a multivariate Gaussian distribution. The proposed methodology and proof techniques can be extended to distributions in which conditional expectations are fully determined by the mean and variance, such as certain exponential family distributions. For heavy-tailed or non-Gaussian distributions, it may be possible to bound causal effects; however, to achieve full identification, we impose these stricter assumptions to ensure that the theoretical guarantees hold.

Our identification result in Theorem 1 is applicable to a wide class of confounded ANMs where a multivariate additive Gaussian noise affects all variables. We explain with a simple example that while non-parametric identification will fail, our inference scheme will still be able to identify the average causal effect under our parametric assumption. Consider a causal graph on variables $X_1, X_2, Y$ of the form $X_1 \rightarrow Y \leftarrow X_2$ with a latent variable $U \sim \text{Ber}(0.5)$ pointing to all of the variables. Consider two SCMs (Structural Causal Models) consistent with the above causal graph: $\mathcal{M}_1$, with structural equations $X_1 = X_2 = U$ and $Y = (X_1 - X_2)(1 - U)$; and a second one, $\mathcal{M}_2$, with $X_1 = X_2 = U$ and $Y = (X_1 - X_2)(U)$. It can be easily verified that both SCMs entail the same observational distribution and interventional distribution $P[Y \mid do(X_1 = x_1, X_2 = x_2)]$. However, they differ on the $do(X_1 = 0)$ intervention. For $\mathcal{M}_1$, we have $\mathbb{E}[Y \mid do(X_1 = 0)] = \mathbb{E}[-(U)(1 - U)] = 0$, but for $\mathcal{M}_2$, we have $\mathbb{E}[Y \mid do(X_1 = 0)] = \mathbb{E}[-U^2] = -0.5$. Thus, $\mathbb{E}[Y \mid do(X_1)]$ is not identifiable from the observational distribution and $P[Y \mid do(X_1, X_2)]$. However, note that operating under our parametric assumptions on the underlying SCM, we can identify all the Average Causal Effects from just the observational distribution and the interventional distribution $P[Y \mid do(X_1, X_2)]$ as implied by Theorem 1. Note that $Pa(X_1) = Pa(X_2) = \emptyset$ for this case.

In a previous study by Jeunen et al. (2022), it was demonstrated that the combination of the observational distribution and interventional distribution ($P[Y|do(\mathbf{X})]$) enables the identification of all AOIs within the framework of confounded ANMs when the treatments have no causal effects on each other. Furthermore, they assume the Gaussian noise to have a zero mean. However, in a more general scenario where treatments can have causal effects, interventions on all treatments and the observational distributions are insufficient to identify all AOIs. In this case, only the AOI for the sink treatment can be identified Jeunen et al. (2022). A sink treatment is one that does not cause any treatments; for example, $X_4$ is the sink treatment in causal graphs $G_1$ and $G_2$ in Figure 1. Thus, to identify all possible AOIs for $G_1$ and $G_2$, we require access to more interventional data, as specified by identification theorem 1. Also, the identification result in Jeunen et al. (2022) for the case where treatments do not cause each other can be derived from Theorem 1, since here $Pa(X_i) = \emptyset$ for all $i$, which implies that the observational distribution and interventions on all treatments are sufficient to identify all AOIs. Thus, Theorem 1 generalizes the identification results presented in Jeunen et al. (2022).

The Theorem 1 can be useful for ANMs with a large number of treatments because the size of the intervention set will be much smaller as compared to the total number of possible subsets of the treatment set $\mathbf{X}$. For instance, consider an ANM with ten treatments and one outcome. There will be $2^{10} = 1024$ possible AOIs of the form $\mathbb{E}[Y|do(\mathbf{W})]$ where $\mathbf{W} \subseteq \mathbf{X}$. The number of sufficient interventional data sets for inference is at most linear in the number of variables of the graph. This implies that for ANMs with large number of treatments, we can identify an exponentially large number of AOIs using a much smaller number of interventional data sets. There are 16 possible AOIs for all causal graphs shown in Figure 1. Consider the graph structure $G_1$ given in the Figure 1(a). Following the Theorem 1, the intervention target set $\mathcal{I} = \{\emptyset, \{X_1, X_2, X_3, X_4\}, \{X_2\}, \{X_1, X_2, X_3\}\}$. Thus we can identify all possible 16 AOIs using the four interventional data sets. Similarly, for the causal graphs in the Figure 1(b) and Figure 1(c), the intervention target sets are: $\{\emptyset, \{X_1, X_2, X_3, X_4\}, \{X_1\}, \{X_2\}, \{X_3\}\}$ for $G_2$, and $\{\emptyset, \{X_1, X_2, X_3, X_4\}\}$ for $G_3$. There is a possible extension of the result in Theorem 1 to the case when additive noise for a particular treatment variable say $X_a$ is independent of the additive noise for other treatments and the outcome variable. In this case, we can drop some intervention targets from the set $\mathcal{I}$. The formal statement of this result is presented in the corollary 1.

**Corollary 1** *Let $\langle \{\boldsymbol{X}, Y\}, \boldsymbol{U}, \mathcal{F}, P_{\boldsymbol{u}} \rangle$ be an SCM Where $X_i = f_i(Pa(X_i)) + U_i$, $Y = f_Y(\boldsymbol{X}) + U_Y$ and $P_{\boldsymbol{u}} \sim \mathcal{N}(\mu, \Sigma)$ and suppose we have $U_a \perp\!\!\!\perp U_b \ \forall b \neq a$ for some $a \in [n]$. In this setting, any possible AOI is identifiable from the conjunction of the two data regimes; the observational distribution and interventional distributions that includes $P[Y|do(\boldsymbol{X})]$ and $\{P[\boldsymbol{V} \setminus Pa(X_i)|do(Pa(X_i))], \ \forall i \neq a\}$.*

The scope of Theorem 1 is not limited to a specific graph structure, where there are $n$ treatments and one outcome variable. The established identifiability result can be generalized to other causal graph structures, provided that the noise or latent variables satisfy the conditions of being additive, and jointly Gaussian. For example, consider an SCM $\mathcal{M} = \langle \mathbf{X}, \mathbf{U}, \mathcal{F}, P_{\mathbf{u}} \rangle$ where $X_i = f_{X_i}(Pa(X_i)) + U_{X_i}$. Let $X_i \in \mathbf{X}$ and $\mathbf{W} \subseteq \mathbf{X}$ such that $X_i \notin \mathbf{W}$. Also assume $\mathbf{T} = Pa(X_i) \setminus \mathbf{W}$ where $Pa(X_i)$ is the parent set of the variable $X_i$ in the SCM $\mathcal{M}$. The conditional AOI of the form $\mathbb{E}[X_i|do(\mathbf{W}), \mathbf{T}]$ can be identified from the conjunction of the interventional distributions $do(Pa(X_i))$ and $do(Pa(T))$. We have $\mathbb{E}[X_i|do(\mathbf{W}), \mathbf{T}] = f_{X_i}(Pa(X_i)) + \mathbb{E}[U_{X_i}|do(\mathbf{W}), \mathbf{T}]$. The structural equation $f_{X_i}$ and the conditional expectation of the noise can be identified using the aforementioned interventional distributions. Theorem 1 provides the sufficient conditions to identify all the AOIs in ANMs with Gaussian noise. We extend Theorem 1 to give necessary and sufficient conditions to identify all the AOIs in confounded ANMs in Theorem 2.

**Theorem 2 (Necessary & Sufficient Conditions for AOIs Identifiability in Confounded ANMs)** *Let $\langle \boldsymbol{V} = \{\boldsymbol{X}, Y\}, \boldsymbol{U}, \mathcal{F}, P_{\boldsymbol{u}} \rangle$ be an SCM where $X_i = f_i(Pa(X_i)) + U_i$, $Y = f_Y(\boldsymbol{X}) + U_Y$ and $P_{\boldsymbol{u}} \sim \mathcal{N}(\mu, \Sigma)$. All estimands of the form $\mathbb{E}[Y|do(\boldsymbol{W})]$ where $\boldsymbol{W} \subseteq \boldsymbol{X}$ are identifiable if and only if we have access to interventional distribution $P[Y|do(\boldsymbol{X})]$ and the interventional distributions $P[\boldsymbol{V} \setminus \boldsymbol{S}_i|do(\boldsymbol{S}_i)]$ such that $X_i \notin \boldsymbol{S}_i$ and $Pa(X_i) \subseteq \boldsymbol{S}_i$ for every treatment $X_i \in \boldsymbol{X}$.*

Although it appears as if Theorem 2 states that, in the case of general confounded ANMs with $n$ treatments, $\mathcal{O}(n)$ interventional datasets are required to identify all possible AOIs, we show that it is possible to reduce this to a poly-logarithmic order by using a randomized algorithm to select intervention targets similar to the one proposed in Kocaoglu et al. (2017). We also demonstrate that this set of interventions is adequate for learning the observable causal graph, i.e., the induced graph between observed variables.

## 4.2 A Randomized Algorithm for Identifying AOIs

A deterministic approach to learning the observable graph will require $n$ interventions (Addanki & Kasiviswanathan, 2021). The observable graph is the induced subgraph of the original causal graph on the observed variables only, containing directed edges between observed variables while omitting edges involving unobserved (latent) variables. For every observed variable $X_i$, the post-interventional graph $G_{\overline{\mathbf{X} \setminus X_i}}$ is constructed. A pair of observed variable $X_j$ and $X_i$ will be d-connected in $G_{\overline{\mathbf{X} \setminus X_i}}$ only when $X_j \in Pa(X_i)$. This approach enables thefn learning of the observable graph using $n$ interventions, each involving $n-1$ variables. Compared to a deterministic approach, which would require $n$ interventions, our randomized algorithm requires fewer interventions, i.e., $\mathcal{O}(8\alpha \ d_{\max} \ \log^2(n))$. Additionally, it returns the necessary interventional datasets for inference with a probability of at least $1 - \frac{1}{n^{\alpha-1}}$. By utilizing ancestral graphs and incorporating random interventions, the algorithm learns the observable graph and generates sufficient interventional datasets for inference purposes with high probability.

**Definition 2** *A collection of subsets $\mathcal{S} = \{\boldsymbol{S}_1, \boldsymbol{S}_2, ..., \boldsymbol{S}_m\}$ of the treatment variable set $\boldsymbol{X}$ is a **strongly separating set system** if for any pair of variables $X_i$ and $X_j$, there exists $\boldsymbol{S}_i$ and $\boldsymbol{S}_j$ such that $X_i \in \boldsymbol{S}_i \setminus \boldsymbol{S}_j$ and $X_j \in \boldsymbol{S}_j \setminus \boldsymbol{S}_i$.*

We can construct a strongly separating using the binary expansion of numbers from 1 to $n$. Specifically, we consider numbers of length $\lceil \log(n) \rceil$ in binary representation. For each bit $i$, we create a set $\mathbf{S}_i$ that includes the numbers where the $i$-th bit is 1, and another set $\mathbf{S}_i'$ that includes the numbers where the $i$-th bit is 0. The family of all such sets forms a strongly separating system (Kocaoglu et al., 2017). This implies that we can always find a strongly separating set system $\mathcal{S}$ on the ground set $\mathbf{X}$ with $|\mathcal{S}| \leq 2\lceil \log(n) \rceil$. We can learn all the ancestral relations in a causal graph with a maximum of $2\lceil \log(n) \rceil$ interventions using the separating system $\mathcal{S}$.

---

**Algorithm 1:** Learn the transitive closure of the graph given access to CI-testing and query access to sample causal model under any intervention i.e. $\mathcal{M}_{do(\mathbf{S}_i)}$.

---

**Function** `LearnTransitiveClosure`($\mathcal{M}$):

    $\mathbf{E} = \emptyset$

    Construct $\mathcal{S} = \{\mathbf{S}_1, \mathbf{S}_2, ..., \mathbf{S}_m\}$ where m $\leq 2\lceil \log(n) \rceil$ as a strongly separating set system

    **for** $i = 1 : 2\lceil\log(n)\rceil$ **do**

        **for** *every pair* $X_i \in \boldsymbol{S}_i$ *and* $X_j \in \boldsymbol{V} \setminus \boldsymbol{S}_i$ **do**

            Use samples from $\mathcal{M}_{do(\mathbf{S}_i)}$ for CI test.

            **if** $(X_j \not\perp\!\!\!\perp X_i)_{G_{\overline{S_i}}}$ **then**

                $\mathbf{E} \longleftarrow \mathbf{E} \cup (X_j, X_i)$

    **return** The graph's transitive closure $(\mathbf{V}, \mathbf{E})$.

**End Function**

---

**Algorithm 2:** Learn the observable graph: Accepts two parameters $\alpha$ and maximum graph degree $d_{max}$ and outputs the observable sub-graph and sufficient interventional data sets for inference.

---

**Function** `LearnObservableGraph`($\mathcal{M}$):

    $\mathbf{E} = \emptyset$

    $\mathcal{I}Data = $ Samples from observational distribution & intervention on all treatments i.e. $do(\mathbf{X})$.

    **for** $i = 1 : 4\alpha \, d_{max} \, \log(n)$ **do**

        $\mathbf{S} = \emptyset$

        **for** $X_i \in \boldsymbol{X}$ **do**

            $\mathbf{S} \leftarrow \mathbf{S} \cup X_i$ with probability $1 - \frac{1}{d_{max}}$

        $G_{\overline{\mathbf{S}}}^{tc} = LearnTransitiveClosure(\mathcal{M}_{do(\mathbf{S})})$

        Compute the transitive reduction $Tr(G_{\overline{\mathbf{S}}}^{tc})$ & add any missing edges from $Tr(G_{\overline{\mathbf{S}}}^{tc})$ to $\mathbf{E}$

        Intervene and sample data from $\mathcal{M}_{do(S)}$ & $\mathcal{I}Data \leftarrow \mathcal{I}Data \cup$ Data from $\mathcal{M}_{do(S)}$

    **return** The observable graph structure $(\mathbf{V}, \mathbf{E})$ and interventional data samples in $\mathcal{I}Data$.

**End Function**

---

**Definition 3** *For a causal graph $G = (\boldsymbol{V}, \boldsymbol{E})$, we define a new DAG $G^{tc} = (\boldsymbol{V}, \boldsymbol{E}^{tc})$ such that, for any pair of vertices $V_i$ and $V_j \in \boldsymbol{V}$, the directed edge $V_i \to V_j$ is included in $G^{tc}$ only when there is a directed path from $V_i$ to $V_j$ in $G$. We refer to this new DAG, $G^{tc}$, representing ancestral relations in $G$ as the **transitive closure** of $G$.*

Algorithm 1 can be employed to learn the transitive closure of the causal graph $G^{tc}$ corresponding to the SCM $\mathcal{M}$ with $2\lceil\log(n)\rceil$ interventions (Kocaoglu et al., 2017).

**Definition 4** *Consider a DAG $G = (\boldsymbol{V}, \boldsymbol{E})$ with its transitive closure $G^{tc}$. The **transitive reduction** of $G$, denoted by $Tr(G) = (\boldsymbol{V}, \boldsymbol{E}_r)$, is a DAG with the minimum number of edges that has the same transitive closure as $G^{tc}$.*

The transitive reduction of an acyclic graph is unique and can be computed in polynomial time (Aho et al., 1972). Furthermore the set of directed edges $E_r$ in $Tr(G)$ is the subset of directed edges in $G$, i.e., $E_r \subseteq E$ (Aho et al., 1972). Algorithm 2 randomly samples the intervention target sets $\mathbf{S}$ and accumulates the edges learned from the transitive closure of the post-interventional graphs to finally learn the observable graph structure. We demonstrate that Algorithm 2 not only learns the observable graph but also returns the required interventional datasets for inference with a high probability in just polylogarithmic number of interventions (Theorem 3). In order to ensure the identifiability of all AOIs using Theorem 1, we need access to the observational distribution and interventional distributions, which include $P[Y|do(\mathbf{X})]$ and $P[\mathbf{V} \setminus Pa(X_i)|do(Pa(X_i))]$ for all $i = 1$ to $n$. This implies that $\mathcal{O}(n)$ interventional data sets are required to infer all AOIs. However, to demonstrate that a polylogarithmic number of interventional data sets from Algorithm 2 is sufficient to infer all AOIs, we use a more general criterion that the interventional data sets must satisfy to ensure the identification of all AOIs, as stated in Theorem 2. Once we have access to all the

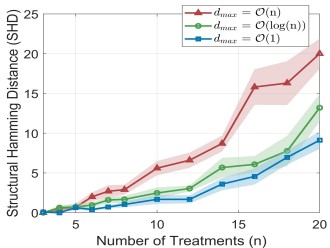
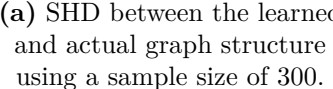
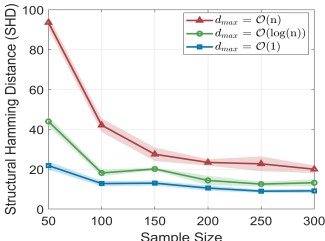
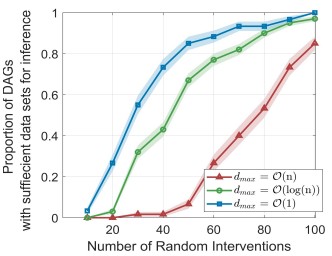

**(a)** SHD between the learned and actual graph structure using a sample size of 300.

**(b)** SHD between the learned and actual graph structure vs. sample size ($n = 20$).

**(c)** Proportion of DAGs with sufficient data sets for inference vs. interventions ($n = 20$).

Figure 2: Simulation results for the proposed randomized algorithm for ANMs with a small number of treatments.

interventional datasets that meet the criteria mentioned in Theorem 2, we can compute all possible AOIs. Algorithm 2 takes two input parameters: $d_{\max}$ and $\alpha$. The algorithm requires a polylogarithmic number of interventional data sets, specifically $\mathcal{O}(\log^3(n))$ when $\alpha = \log(n)$. Note that the required number of interventional data sets scales linearly with the maximum graph degree ($d_{\max}$).

**Theorem 3** *Suppose that $d_{max}$ is greater or equal to the maximum degree in the observable graph of the ANM. Algorithm 2 requires a maximum of $8\alpha\, d_{max}\, \log^2(n)$ interventions to learn the true observable graph with a probability of at least $1 - \frac{1}{n^{\alpha-2}}$. The algorithm also returns the required interventional data sets to compute the noise variance matrix and the AOIs with a success probability of $1 - \frac{1}{n^{\alpha-1}}$.*

**Corollary 2** *If the parameter $\alpha$ is chosen as $\log(n)$ in Algorithm 2, the maximum number of required interventions will be $8d_{max}\, \log^3(n)$ to return the observable graph with a probability of at least $1 - \frac{1}{n^{\log(n)-2}}$. The algorithm will also return required interventional data sets for inference with a success probability of $1 - \frac{1}{n^{\log(n)-1}}$.*

The proof of Theorem 3 is provided in the supplementary material. The randomized algorithm not only provides sufficient datasets for inference but also enables learning the underlying graph structure in ANMs when it is unknown. Note that we do not impose a bound on the maximum number of nodes in an intervention set, allowing the total number of experiments to remain poly-logarithmic. If one wishes to restrict the size of each intervention set, an $(n, k)$ separating system, where the size of each intervention set is bounded by $k$, from Theorem 1 Shanmugam et al. (2015) can be used; however, in that case, the total number of experiments may no longer remain poly-logarithmic and may become linear in the number of nodes in the worst case.

## 5 Empirical Validation

We empirically validate our method, which includes experiments on synthetic and semi-synthetic data based on the HEALTHCARE Bayesian network from the bnlearn library repository.

### 5.1 Synthetic Experiment

In this section, we evaluate our combined discovery and inference scheme using synthetic data. Our goal is to evaluate the algorithm's ability to accurately determine the graph structure, generate sufficient interventional data sets for inference, and provide precise estimates of AOIs. We evaluate the combined causal discovery algorithm and inference by randomly generating DAGs with sets of treatments $\mathbf{X}$, which all affect the outcome $Y$. A multivariate Gaussian noise is added to the structural equations of the treatments and outcomes. We test different sizes of treatment sets, with $n$ ranging from 3 to 20, and generate 50 random DAGs for each size. The curse of dimensionality presents a challenge in testing conditional independence for continuous variables. We address testing conditional independence for continuous variables using an kernel matrix-based method that outperforms discretization-based approaches in terms of accuracy (Zhang et al., 2012).

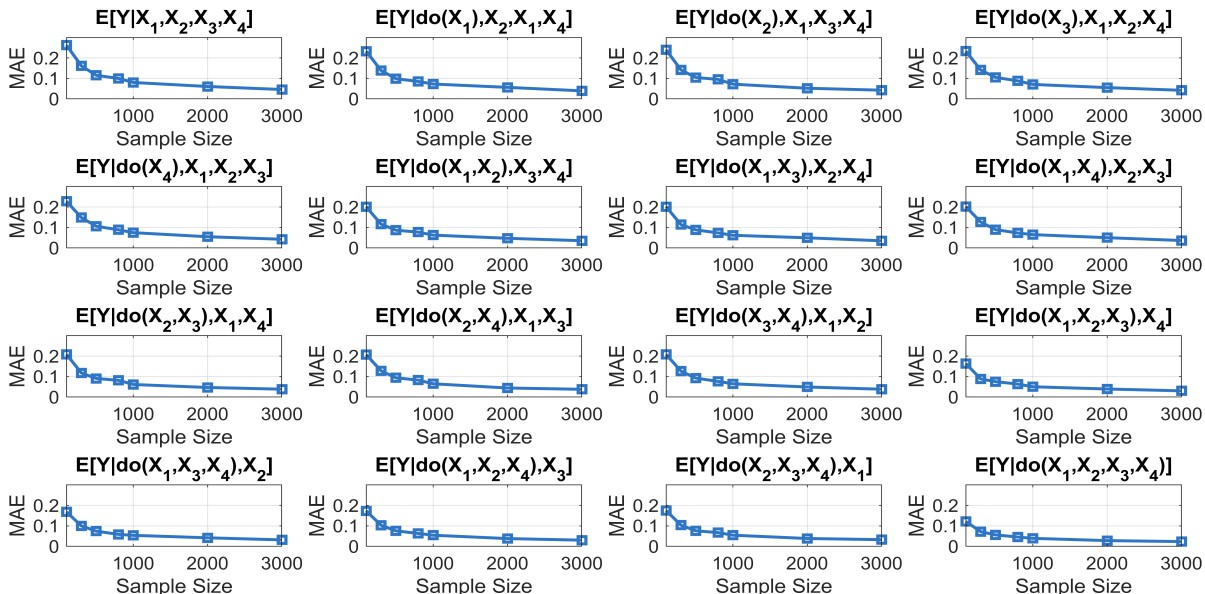

Figure 3: Mean absolute error (MAE) in estimating AOIs under various interventions for randomly generated ANMs with 4 treatment variables, across different sample sizes.

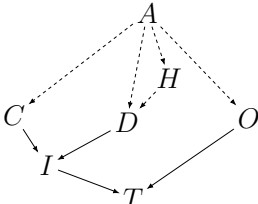

Figure 4: Graph for HEALTHCARE Bayesian Network

Figure 2(a) plots the Structural Hamming Distance (SHD) for randomly generated DAGs against the number of treatments. The results show that the SHD increases along with the increase in the number of treatments, but at a slower pace compared to the increase in the number of possible edges in the graph. This implies that the discovery algorithm being used performs well in determining the true structure of the DAGs, even when the number of treatments increases. Figure 2(b) plots sample size versus the performance of the discovery algorithm. The performance is evaluated by the SHD between the learned and true DAGs. The results demonstrate that the SHD decreases as the number of samples increases. Figure 2(c) presents the relationship between the number of random interventions and the proportion of DAGs that have sufficient data for inference. The results are based on 50 randomly generated DAGs, each with $n = 20$ treatments. The plot confirms that as the number of interventions increases, the probability of having sufficient data for inference also increases. This indicates that a larger number of interventions can enhance the chances of obtaining adequate data for causal inference. Furthermore, the results suggest that the number of interventions required to obtain sufficient data for inference increases with the graph's maximum degree.

To demonstrate the performance of our inference scheme, we apply our combined discovery and inference algorithm to 50 randomly generated ANMs, each with a fixed number of treatments ($n = 4$). We analyze and plot the error in approximating each of the 16 AOIs against the sample size in Figure 3.

## 5.2 Semi-synthetic Experiment Based on HEALTHCARE Bayesian Network

In order to demonstrate the effectiveness of our inference scheme, we evaluate it through a semi-synthetic experiment using the HEALTHCARE Bayesian network from bnlearn repository (Scutari, 2009). The corresponding causal graph is shown in Figure 4. The variables $A$ and $H$ are discrete, while the remaining

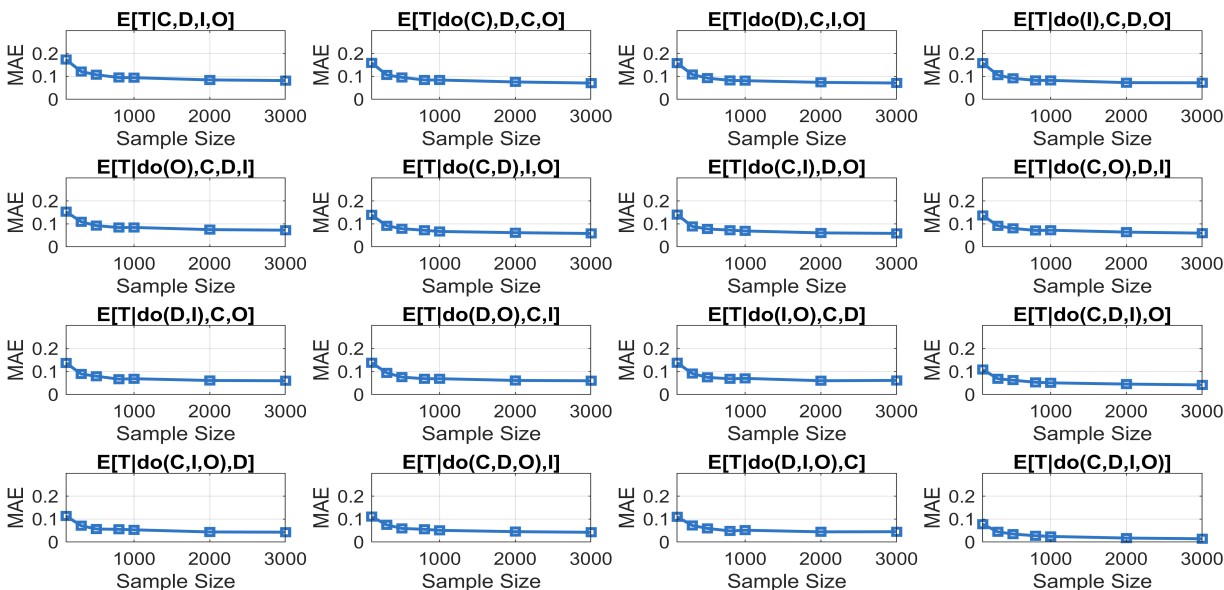

Figure 5: Mean absolute error (MAE) in estimating AOIs under various interventions for HEALTHCARE Bayesian Network.

variables are continuous. The conditional distribution of variables $C$, $D$, $O$, $I$ and $T$ given their parents follows a Gaussian distribution with a specific variance and mean determined by their parent variables' values. For instance, the conditional probability distribution of $I$ given its parents $C$ and $D$ is $\mathcal{N}(100d, \sigma_c^2)$. The variance $(\sigma_c^2)$ of the distribution depends on the realization $(c)$ of the discrete variable $(C)$. To introduce confounding into the network, we assume that we do not observe the variables $A$ and $H$, which now act as latent confounders, as indicated by the dashed edges in Figure 4. The outcome variable is $T$, and the treatment variables include $C$, $D$, $O$, $I$. We apply our inference scheme to estimate all 16 possible AOIs using the three interventional datasets ( $\mathcal{I} = \{\phi, \{C, D\}, \{C, D, O, I\}\}$ ), and we plot the corresponding estimation errors in Figure 5. As our assumption of additive Gaussian noise remains unsatisfied, the estimation errors are higher when compared to the synthetic experiment. Nevertheless, the central finding remains that, even when the Gaussianity assumption is unmet, our inference scheme yields reasonably precise estimates. The results of our experiment can be found at `https://github.com/QasimElahi/Code-for-TMLR-paper-Identification-in-Confounded-Additive-Noise-Models`.

# 6    Conclusion

We propose a scheme to estimate the average outcome under various interventions for confounded additive noise models. Our approach demonstrates that a relatively small number of interventional distributions is sufficient to determine all possible AOIs in ANMs. We also show that the interventional datasets characterized by our method are necessary to perform inference for all possible AOIs in confounded ANMs. By employing a randomized algorithm, we significantly reduce the number of interventions required, achieving a poly-logarithmic scale relative to the number of treatments. Through synthetic and semi-synthetic experiments, we validate the reliability and utility of our approach for both causal discovery and inference. These results highlight the practical feasibility of our method for learning observable causal structures and performing efficient inference even in the presence of latent confounders.

## Acknowledgment

This research has been supported in part by NSF CAREER 2239375, IIS 2348717, Amazon Research Award, Adobe Research and Intuit.

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

# A  Supplementary Material

In the following subsections, we provide comprehensive and formal mathematical proofs for the the theorems presented in the main paper.

## A.1  Pearl's Rules of do-Calculus (Pearl (2009))

Let $G$ represent the causal DAG, and let $P$ denote the probability distribution induced by the corresponding causal model. For any disjoint subsets of variables $\mathbf{X}, \mathbf{Y}, \mathbf{Z}$, and $\mathbf{W}$, the following rules apply:

**Rule 1:** (Insertion/deletion of observations): $P(\mathbf{y}|do(\mathbf{x}), \mathbf{z}, \mathbf{w}) = P(\mathbf{y}|do(\mathbf{x}), \mathbf{w})$ if $(\mathbf{Y} \perp\!\!\!\perp \mathbf{Z}|\mathbf{X}, \mathbf{W})_{G_{\overline{\mathbf{x}}}}$.

**Rule 2:** (Action/observation exchange): $P(\mathbf{y}|do(\mathbf{x}), do(\mathbf{z}), \mathbf{w}) = P(\mathbf{y}|do(\mathbf{x}), \mathbf{z}, \mathbf{w})$ if $(\mathbf{Y} \perp\!\!\!\perp \mathbf{Z}|\mathbf{X}, \mathbf{W})_{G_{\overline{\mathbf{x}}\underline{\mathbf{z}}}}$.

**Rule 3:** (Insertion/deletion of actions): $P(\mathbf{y}|do(\mathbf{x}), do(\mathbf{z}), \mathbf{w}) = P(\mathbf{y}|do(\mathbf{x}), \mathbf{w})$ if $(\mathbf{Y} \perp\!\!\!\perp \mathbf{Z}|\mathbf{X}, \mathbf{W})_{G_{\overline{\mathbf{x}}, \overline{\mathbf{z}(\mathbf{w})}}}$ where $\mathbf{Z}(\mathbf{W})$ is the set of nodes in $\mathbf{Z}$ that are not ancestors of any of the nodes in $\mathbf{W}$.

## A.2  Proof of Theorem 1 and Corollary 1

The section provides the proof of the Identifiablity of AOIs in symmetric ANMs. Consider two sets: $\mathbf{X}_{int} \subseteq \mathbf{X}$ of the form $\mathbf{X}_{int} = \{X_{int1}, X_{int2}, ...., X_{int\alpha}\}$ and $\mathbf{X}_O = \mathbf{X} \setminus \mathbf{X}_{int}$ of the form $\mathbf{X}_O = \{X_{O1}, X_{O2}, ...., X_{O\beta}\}$. Also note that $\alpha + \beta = n$.

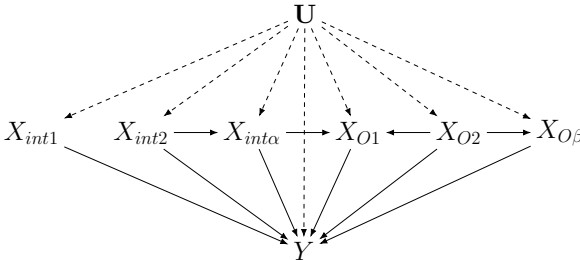

Figure 6: Causal graph ($G$) for a general confounded additive noise model (The edges between the treatments $X_i's$ can be arbitrary so long as no cycle is formed)

$$\mathbb{E}[Y|do(X_{int1} = x_{int1}, ...., X_{int\alpha} = x_{int\alpha}), X_{O1} = x_{O1}, ...., X_{O\beta} = x_{O\beta}] =$$
$$f_Y(x_{int1}, ..., x_{int\alpha}, x_{O1}, ....., x_{O\beta}) \ +$$
$$\mathbb{E}(U_Y|X_{O1} = x_{O1}, ...., X_{O\beta} = x_{O\beta}, do(X_{int1} = x_{int1}, ...., X_{int\alpha} = x_{int\alpha})) \quad (3)$$

From the intervention on all treatments, we have access to the following:

$$\mathbb{E}[Y|do(X_{int1} = x_{int1}, ...., X_{int\alpha} = x_{int\alpha}, X_{O1} = x_{O1}, ...., X_{O\beta} = x_{O\beta})] =$$
$$\mathbb{E}[U_y] + f_Y(x_{int1}, ..., x_{int\alpha}, x_{O1}, ....., x_{O\beta}) \quad (4)$$

The equation 5 gives us the conditional expectation of $U_Y$ given the observed treatment variables.

$$\mathbb{E}[U_Y|X_{O1} = x_{O1}, ...., X_{O\beta} = x_{O\beta}, do(X_{int1} = x_{int1}, ...., X_{int\alpha} = x_{int\alpha})]$$
$$= \mathbb{E}[U_Y|U_{O1} = x_{O1} - f_{O1}(Pa(X_{O1})), ......, U_{O\beta} = x_{O\beta} - f_{O\beta}(Pa(X_{O\beta}))] \quad (5)$$

Where the assignment $Pa(X_{Oi})$ is consistent with the causal query specified in equation 3. In order to show how equation 3 holds, we use the following approach:

$$\mathbb{E}[U_Y | U_{O1} = x_{O1} - f_{O1}(Pa(X_{O1})), ......, U_{O\beta} = x_{O\beta} - f_{O\beta}(Pa(X_{O\beta}))] =$$
$$\mathbb{E}[U_Y | U_{O1} = x_{O1} - f_{O1}(Pa(X_{O1})), ......, U_{O\beta} = x_{O\beta} - f_{O\beta}(Pa(X_{O\beta})), do(X_{int1} = x_{int1}, ...., X_{int\alpha} = x_{int\alpha})] \tag{6}$$

$$\mathbb{E}[U_Y | U_{O1} = x_{O1} - f_{O1}(Pa(X_{O1})), ......, U_{O\beta} = x_{O\beta} - f_{O\beta}(Pa(X_{O\beta}))] =$$
$$\mathbb{E}[U_Y | U_{O1} = x_{O1} - f_{O1}(Pa(X_{O1})), ......, U_{O\beta} = x_{O\beta} - f_{O\beta}(Pa(X_{O\beta})), X_{O1} = x_{O1}, ......$$
$$......, X_{O\beta} = x_{O\beta}, do(X_{int1} = x_{int1}, ...., X_{int\alpha} = x_{int\alpha})]$$
$$= \mathbb{E}[U_Y | X_{O1} = x_{O1}, ...., X_{O\beta} = x_{O\beta}, do(X_{int1} = x_{int1}, ...., X_{int\alpha} = x_{int\alpha})] \tag{7}$$

The equation 6 holds because of Pearl's do-calculus rule 3 since $U_y \perp\!\!\!\perp \mathbf{X}_{int} | \mathbf{U}_O$ in the post interventional graph where incoming edges to the set $\mathbf{X}_{int}$ are removed i.e. $G_{\overline{\mathbf{X}_{int}}}$. Note that $\mathbf{U}_O = [U_{O1}, U_{O2}, \ldots, U_{O\beta}]$ and none of variables in the treatment set can be ancestors or parents of the noise variables. The equation 7 holds because of Pearl's do-calculus rule 1 since $U_y \perp\!\!\!\perp \mathbf{X}_O | \mathbf{U}_O, \mathbf{X}_{int}$ in the post interventional graph where incoming edges to the set $\mathbf{X}_{int}$ are removed, i.e., $G_{\overline{\mathbf{X}_{int}}}$. Now, using the property of conditional expectation of correlated Gaussian random variables, the equation 5 can be rewritten as:

$$\mathbb{E}[U_Y | X_{O1} = x_{O1}, ...., X_{O\beta} = x_{O\beta}, do(X_{int1} = x_{int1}, ...., X_{int\alpha} = x_{int\alpha})] = \mathbb{E}[U_y] + \Sigma_{yx} \Sigma_{xx}^{-1} \mathbf{V},$$

$$\text{where } \Sigma_{yx} = [\, \sigma_{Y(O1)}, \sigma_{Y(O2)}, ....., \sigma_{Y(O\beta)} \,],$$

$$\Sigma_{xx} = \begin{bmatrix} \sigma_{O1}^2 & \sigma_{(O1)(O2)} & \sigma_{(O1)(O3)} & \cdots & \cdots & \cdots & \cdots & \sigma_{(O1)(O\beta)} \\ \sigma_{(O2)(O1)} & \sigma_{O2}^2 & \sigma_{(O2)(O3)} & \ddots & & & & \vdots \\ \vdots & \ddots & & \ddots & \ddots & \ddots & & \vdots \\ \vdots & & & \ddots & \ddots & \ddots & \ddots & \vdots \\ \sigma_{(O\beta)(O1)} & \cdots & & \cdots & \cdots & \cdots & \cdots & \sigma_{(O\beta)}^2 \end{bmatrix},$$

$$\mathbf{V} = [\, x_{O1} - f_{O1}(Pa(X_{O1})) - \mathbb{E}[U_{O1}], \; x_{O2} - f_{O2}(Pa(X_{O2})) - \mathbb{E}[U_{O1}] \,, ....$$
$$....., \; x_{O\beta} - f_{O\beta}(Pa(X_{O\beta})) - \mathbb{E}[U_{O\beta}] \,]^T. \tag{8}$$

In equation 8, the noise parameters $(\Sigma_{yx}, \Sigma_{xx})$ and the structural equations $f_{Oi}(.)$ must be identified from a combination of observational and interventional data. The term $\sigma_{ij}$ denotes the covariance between the noise variables $U_i$ and $U_j$. Our results hold for any noise covariance matrix $\Sigma_{xx}$. However, there may be cases where the covariance matrix is not invertible. For example, in the case of multivariate Gaussian noise, this occurs if and only if at least one component of the multivariate Gaussian noise is a linear combination of other components. In equation 8, we invert the covariance matrix. This step might fail when the covariance matrix is non-invertible. If the noise covariance matrix is not invertible, we can apply Gaussian elimination to identify the dependent noise component(s). These component(s) can then be removed from the conditioning set. Consequently, the corresponding rows of $\Sigma_{xx}$ and the corresponding entries of $\Sigma_{yx}$ can also be removed from equations 8. This ensures that the resulting covariance matrix is invertible, and our identification scheme remains applicable. Thus, a small clarification in our proof is required to address cases where there

are dependent noise components, meaning the covariance matrix is not full rank. Specifically, we need to identify the dependent noise components and remove the corresponding rows from the covariance matrix to ensure it is invertible.

**Identifying the Structural Equations:** We have access to the interventional distributions of the form $Pr[\mathbf{V} \setminus Pa(X_i) \mid do(Pa(X_i))]$. The structural equations $f_{Oi}(.)$, shifted by unknown Gaussian noise means ( $\mathbb{E}[U_{O1}]$ ), can thus be identified by using Equation 9.

$$\mathbb{E}[X_{Oi}|do(Pa(X_{Oi}))] = f_{Oi}(Pa(X_{Oi})) + \mathbb{E}[U_{O1}|do(Pa(X_{Oi}))] = f_{Oi}(Pa(X_{Oi})) + \mathbb{E}[U_{O1}] \tag{9}$$

The Equation 9 holds by Pearl's do-calculus rule 3 since $U_{O1} \perp\!\!\!\perp Pa(X_{Oi})$ in the post interventional graph where incoming edges to the set $Pa(X_{Oi})$ are removed i.e. $G_{\overline{Pa(X_{Oi})}}$.

**Identifying the Noise Co-variances:** After identifying the structural equations, observational samples can be used once more to identify noise samples $U_i$, which are again shifted by unknown noise means, by rearranging the equation $X_i = f_i(Pa(X_i)) + U_i$ as follows:

$$\hat{U}_i = X_i - \mathbb{E}[X_i|do(Pa(X_i))] = X_i - f_i(Pa(X_i)) - \mathbb{E}[U_i] = U_i - \mathbb{E}[U_i] \tag{10}$$

A constant shift in noise values will not affect our noise co-variance estimates. This is because we have $\sigma_{(i)(j)} = \mathbb{E}[(U_i - \mathbb{E}[U_i])(U_j - \mathbb{E}[U_j])] = \mathbb{E}[\hat{U}_i\hat{U}_j]$. We can finally identify the causal effect specified in equation 3 by combining information from equations 4, 8, and 9 as follows:

$$\mathbb{E}[Y|do(X_{int1} = x_{int1}, ...., X_{int\alpha} = x_{int\alpha}), X_{O1} = x_{O1}, ...., X_{O\beta} = x_{O\beta}] =$$
$$\mathbb{E}[Y|do(X_{int1} = x_{int1}, ...., X_{int\alpha} = x_{int\alpha}, X_{O1} = x_{O1}, ...., X_{O\beta} = x_{O\beta})] \quad + \Sigma_{yx}\Sigma_{xx}^{-1}\mathbf{V}$$
$$where \ \ \mathbf{V} = [\ x_{O1} - \mathbb{E}[X_{O1}|do(Pa(X_{O1}))] \ , ....., \ x_{O\beta} - \mathbb{E}[X_{O\beta}|do(Pa(X_{O\beta}))]\ ]^T. \tag{11}$$

This shows that $\mathbb{E}[Y|do(X_{int1} = x_{int1}, ...., X_{int\alpha} = x_{int\alpha}), X_{O1} = x_{O1}, ...., X_{O\beta} = x_{O\beta}]$ is identifiable. Now, since our true objective is to identify the AOI of the form $\mathbb{E}[Y|do(X_{int1} = x_{int1}, ...., X_{int\alpha} = x_{int\alpha})$, we need to marginalize out the extra variables using the interventional distribution $f(X_{O1} = x_{O1}, ...., X_{O\beta} = x_{O\beta}|do(X_{int1} = x_{int1}, ...., X_{int\alpha} = x_{int\alpha}))$, which can be identified as follows:

$$f(X_{O1} = x_{O1}, ...., X_{O\beta} = x_{O\beta}|do(X_{int1} = x_{int1}, ...., X_{int\alpha} = x_{int\alpha})$$
$$= \int \prod_{i=1}^{i=\beta} f(x_{Oi}|pa(X_{Oi}), u_{Oi}) f_{\mathbf{U}}(u_{O1}, u_{O2}, ..., u_{O\beta}) \, d\mathbf{u} \tag{12}$$

Where the assignment $Pa(X_{Oi})$ is consistent with the causal query specified in equation 3. Notice that $X_{Oi} = f_i(Pa(X_{Oi})) + U_{Oi}$ where $f_i(.)$ is a deterministic function. This implies that the probability density function $f(x_{Oi}|pa(X_{Oi}), u_{Oi})$ will just be an impulse centered at $u_{Oi} = x_{Oi} - f_i(Pa(X_{Oi}))$. Thus we have:

$$f(X_{O1} = x_{O1}, ...., X_{O\beta} = x_{O\beta}|do(X_{int1} = x_{int1}, ...., X_{int\alpha} = x_{int\alpha}) =$$
$$f_{\mathbf{U}}(u_{O1} = x_{O1} - f_i(Pa(X_{O1})), u_{O2} = x_{O2} - f_i(Pa(X_{O2})), ..., u_{O\beta} = x_{O\beta} - f_i(Pa(X_{O\beta}))) \tag{13}$$

Where $f_{\mathbf{U}} \sim \mathcal{N}(\mu_x, \Sigma_{xx})$. Let us define $f_{\mathbf{U}'} \sim \mathcal{N}(0, \Sigma_{xx})$. Notice that we cannot directly estimate the structural equation $f_i$ exactly but we identify its shifted version i.e. say $f_i'(.) = f_i + \mathbb{E}[U_i]$ (see equation 9). Thus, instead of the exact $u_{Oi} = x_{Oi} - f_i(Pa(X_{Oi}))$, we have access to $u_{Oi}' = x_{Oi} - f_i'(Pa(X_{Oi})) = x_{Oi} - f(Pa(X_{Oi})) - \mathbb{E}[U_{Oi}] = u_{Oi} - \mathbb{E}[U_{Oi}]$. We can get around this issue by noticing that:

$$f_{\mathbf{U}}(u_{O1}, u_{O2}, ..., u_{O\beta}) = f'_{\mathbf{U}}(u_{O1} - \mathbb{E}[U_{O1}], u_{O2} - \mathbb{E}[U_{O2}], ..., u_{O\beta} - \mathbb{E}[U_{O\beta}]) \tag{14}$$

Thus, we can identify the any desired AOI with access to the interventional distributions of the form $Pr[\mathbf{V} \setminus Pa(X_i) \mid do(Pa(X_i))]$ using the final expression below:

$$\mathbb{E}[Y|do(X_{int1} = x_{int1}, ...., X_{int\alpha} = x_{int\alpha})] =$$
$$\int \mathbb{E}[Y|do(X_{int1} = x_{int1}, ...., X_{int\alpha} = x_{int\alpha}), X_{O1} = x_{O1}, ...., X_{O\beta} = x_{O\beta}]$$
$$f(X_{O1} = x_{O1}, ...., X_{O\beta} = x_{O\beta}|do(X_{int1} = x_{int1}, ...., X_{int\alpha} = x_{int\alpha}) \, d\mathbf{X}_O \tag{15}$$

This completes the proof of Theorem 1. The proof of the corollary 1 just follows from the proof of Theorem 1. There are two possibilities, either $X_a \in \mathbf{X}_{int}$ or $X_a \in \mathbf{X}_O$. In the first case, the interventional distribution of $Pa(X_a)$ is not required to identify AOI given in equation 3. For the second case, we rely on the additional assumption in corollary 1 that $U_a \perp\!\!\!\perp U_b, \forall b \neq a$, we have $\sigma_{Ya} = 0$ and $\sigma_{ba} = 0$. Also, we can use the observational data regime to identify the structural equation of variable $X_a$ which is required to evaluate equation 5. The structural equation $f_a(.)$ can be identified from observational data regime using equation 16:

$$\mathbb{E}[X_a|Pa(X_a)] = f_a(Pa(X_a)) + \mathbb{E}[U_a|Pa(X_a)] = f_a(Pa(X_a)) + \mathbb{E}[U_a] \tag{16}$$

Thus, in the case where there is no edge pointing from latent variable $\mathbf{U}$ to a certain treatment $X_a$, the intervention $do(Pa(X_a))$ is no longer required to identify the structural equation of $X_a$. This concludes the proof of corollary 1.

## A.3 Proof of Theorem 2

We prove the sufficiency and necessity of the conditions for identifiability of AOIs in confounded ANMs with Gaussian noise from Theorem 2 in separate subsections below:

### A.3.1 Proof of Sufficiency:

Theorem 2 states that any estimand of the form $\mathbb{E}[Y|do(\mathbf{W})]$ where $\mathbf{W} \subset \mathbf{X}$ can be identified from a collection of interventional distributions including $P[Y|do(\mathbf{X})]$ and $P[\mathbf{V} \setminus \mathbf{S}_i|do(\mathbf{S}_i)]$, such that $X_i \notin \mathbf{S}_i$ & $Pa(X_i) \subseteq \mathbf{S}_i, \quad \forall i = 1, \ldots, n$. Theorem 1 states that access to the observational distribution, interventions on all treatments, and interventional datasets targeting the parents of each treatment variable is sufficient to identify all AOIs. The only difference is that interventions on parents of treatments $Pa(X_i)$ are replaced by interventions on $\mathbf{S}_i$ such that $X_i \notin \mathbf{S}_i$ & $Pa(X_i) \subseteq \mathbf{S}_i, \quad \forall i = 1, \ldots, n$. The Proof of Sufficiency can be inferred from the proof of Theorem 1, except that we need to show we can still identify the structural equations of the treatment variables shifted by unknown Gaussian noise means using interventional distributions $P[\mathbf{V} \setminus \mathbf{S}_i|do(\mathbf{S}_i)]$ instead of the interventional distribution of the form $P[\mathbf{V} \setminus Pa(X_i)|do(Pa(X_i))]$.

$$\mathbb{E}[X_i|do(\mathbf{S}_i)] = f_i(Pa(X_i)) + \mathbb{E}[U_i|do(\mathbf{S}_i)] = f_{Oi}(Pa(X_{Oi})) + \mathbb{E}[U_{O1}] \tag{17}$$

Equation 17 holds due to the fact that $X_i \notin \mathbf{S}_i$ & $Pa(X_i) \subseteq \mathbf{S}_i$ and by Pearl's do-calculus rule 3, since $U_i \perp\!\!\!\perp \mathbf{S}_i$ in the post-interventional graph where incoming edges to $\mathbf{S}_i$ are removed, i.e., $G_{\overline{\mathbf{S}_i}}$. Thus, we can still identify the structural equations shifted by unknown Gaussian noise means using the interventional distributions $P[\mathbf{V} \setminus \mathbf{S}_i|do(\mathbf{S}_i)]$, such that $X_i \notin \mathbf{S}_i$ & $Pa(X_i) \subseteq \mathbf{S}_i, \quad \forall i = 1, \ldots, n$, and the rest of the identifiability proof follows from the proof of Theorem 1.

### A.3.2 Proof of Necessity:

Consider the ANM $Y = f_Y(\mathbf{X}) + U_Y$ and $X_i = f_i(Pa(X_i)) + U_i$ for all $i \in [n]$, where $U = [U_1, U_2, \ldots, U_n, U_Y]$ is multivariate Gaussian noise. Suppose there exists a treatment $X_j \in \mathbf{X}$ such that we do not have access

to an interventional dataset $\mathbf{S}_j$ for which $Pa(X_j) \subseteq \mathbf{S}_j$ and $X_j \notin \mathbf{S}_j$, while we have access to all other possible interventional data. Using the interventional data of the form $do(\mathbf{S}_i = \mathbf{s}_i)$ and $do(\mathbf{S}_i = \mathbf{s}'_i)$ for every treatment $X_i$ such that $i \neq j$, where $Pa(X_i) \subseteq \mathbf{S}_i$ and $X_i \notin \mathbf{S}_i$, we have the following:

$$\mathbb{E}[X_i \mid do(\mathbf{S}_i = \mathbf{s}_i)] = f_i(pa(X_i)) + \mathbb{E}[U_i] \tag{18}$$

$$\mathbb{E}[X_i \mid do(\mathbf{S}_i = \mathbf{s}'_i)] = f_i(pa'(X_i)) + \mathbb{E}[U_i] \tag{19}$$

Using the above equations, we can identify the noise mean as well as the structural equation $f_i$. We can also rearrange the equation to recover the noise samples and identify the noise distributions. In fact, given access to interventional datasets $\mathbf{S}_i$ such that $Pa(X_i) \subseteq \mathbf{S}_i$ and $X_i \notin \mathbf{S}_i$ for every treatment $X_i \in \mathbf{X} \setminus \{X_j\}$, we can identify the joint distribution of the noise variables $\{U_1, U_2, \dots, U_{j-1}, U_{j+1}, \dots, U_n, U_Y\}$ and all the structural equations except for $f_j$. The parameters that can differ across the two ANMs are the equation $f_j$ and the noise distribution parameters $\mathbb{E}[U_j]$, $\sigma_{jj}$, and $\sigma_{ij}$ for all $i \neq j$.

For the treatment $X_j$, we don't have access to an interventional dataset $S_j$ such that $Pa(X_j) \subseteq S_j$ and $X_j \notin S_j$. Now, we want to construct two ANMs, (1) and (2), which have different interventional distributions for an intervention $S_j$ such that $Pa(X_j) \subseteq S_j$ and $X_j \notin S_j$, but agree on all other interventional distributions in the collection.

$$X_j^{(1)} = f_j^{(1)}(Pa(X_j)) + U_j^{(1)} \tag{20}$$

$$X_j^{(2)} = f_j^{(2)}(Pa(X_j)) + U_j^{(2)} \tag{21}$$

$$Y^{(1)} = \sum_{X_i^{(1)} \in \mathbf{X}} X_i^{(1)} + U_Y^{(1)} \tag{22}$$

$$Y^{(2)} = \sum_{X_i^{(2)} \in \mathbf{X}} X_i^{(2)} + U_Y^{(2)} \tag{23}$$

Suppose that the functions $f_j^{(1)}(\cdot)$ and $f_j^{(2)}(\cdot)$ are equal everywhere except at $Pa(X_j) = \boldsymbol{\beta}$. One possible choice is $f_j^{(1)}(Pa(X_j)) = a \times \mathbb{1}\{Pa(X_j) = \boldsymbol{\beta}\}$ and $f_j^{(2)}(Pa(X_j)) = b \times \mathbb{1}\{Pa(X_j) = \boldsymbol{\beta}\}$, where $a \neq b$. Additionally, assume that the multivariate Gaussian noise distribution is the same across the ANMs. This implies that the distribution of $X_j$ is different across the two ANMs under the intervention $do(\boldsymbol{S}_j = \boldsymbol{s}_j)$, where $Pa(X_j) = \boldsymbol{\beta}$. Therefore, the two ANMs have different interventional distributions for $do(\boldsymbol{S}_j)$. Note that $\boldsymbol{S}_j$ is any subset of the treatment set $\boldsymbol{X}$ such that $Pa(X_j) \subseteq \boldsymbol{S}_j$ and $X_j \notin \boldsymbol{S}_j$. Given that the distribution of $X_j$ is different across the two ANMs under some intervention $do(\boldsymbol{S}_j = \boldsymbol{s}_j)$, and that both ANMs share the same noise distribution and structural equations $f_j$ for all $j \neq i$, the AOIs $\mathbb{E}[Y^{(1)} \mid do(\boldsymbol{S}_j = \boldsymbol{s}_j)]$ and $\mathbb{E}[Y^{(2)} \mid do(\boldsymbol{S}_j = \boldsymbol{s}_j)]$ will be different, assuming that both ANMs have the same structural equation for the outcome $f_Y(\mathbf{X}) = \sum_{X_i \in \mathbf{X}} X_i$ as in equations 22 and 23.

We still need to show that the ANMs will have the same interventional distributions other than $\mathbf{S}_j$, where $X_j$ itself is not intervened on. Under any intervention other than $\mathbf{S}_j$ where $X_j$ itself is not intervened on, the probability that $Pa(X_j) = \boldsymbol{\beta}$ is almost surely zero because of the additive Gaussian noise in the structural equations. Thus, the two ANMs constructed will almost surely have the same interventional distributions for all the target sets except $\mathbf{S}_j$ where $\mathbf{S}_j \subseteq \mathbf{X}$ such that $Pa(X_j) \subseteq \mathbf{S}_j$ and $X_j \notin \mathbf{S}_j$. Thus, the given two ANMs can only be distinguished using the intervention on $\mathbf{S}_j$. This proves that all AOIs can't be identified unless we have access to an interventional dataset $\mathbf{S}_i$ for every treatment $X_i \in \mathbf{X}$, such that $Pa(X_i) \subseteq \mathbf{S}_i$ and $X_i \notin \mathbf{S}_i$. This concludes the proof of the necessary conditions for the identifiability of AOIs in ANMs with Gaussian noise, as stated in Theorem 2.

## A.4 Proof of Theorem 3

Recall that in mathematics, specifically in order theory, a partial order on a set is an arrangement where certain elements have a defined precedence over others. The term "partial" indicates that not every pair of elements needs to be comparable in terms of their ordering. In order to prove Theorem 3 we rely on the following lemma from Kocaoglu et al. (2017).

**Lemma 1** *Kocaoglu et al. (2017) Consider a graph $G$ with observable variables $\boldsymbol{X}$ and an intervention set $S \subseteq \boldsymbol{X}$. Consider the post-interventional observable graph $G_{\overline{S}}$ and a variable $X_i \in \boldsymbol{X} \setminus S$. Let $V \in Pa(X_i)$ be such that all the parents of $X_i$ above $V$ in partial order are included in the intervention set $S$. This implies that $\{W : \sigma(W) > \sigma(V), W \in Pa(X_i)\} \subseteq S$ [1]. Then, the directed edge $(V, X_i) \in E(Tr(G_{\overline{S}}))$. The properties of transitive reduction yields $Tr(G_{\overline{S}}) = Tr(G_{\overline{S}}^{tc})$. Consequently, the transitive reduction of $G_{\overline{S}}^{tc}$ may be used to learn the directed edge $(V, X_i)$.*
*(Note: $E(G)$ denotes the edges of the DAG $G$).*

The proof of Theorem 3 is split into two parts. In the first part, we show that the probability of learning the true graph is $1 - \frac{1}{n^{\alpha-2}}$. In the second part, we show that the probability of having sufficient interventional data sets for inference is $1 - \frac{1}{n^{\alpha-1}}$. Consider a directed edge $(V, X_i)$ in graph $G$. Assume that the number of the direct parents of $X_i$ above $V$ is $d_i$ where $d_i \leq d_{max}$. Let $\mathcal{E}_i(V)$ be the following event: $X_i \notin S$ & $\{W : \sigma(W) > \sigma(V), W \in Pa(X_i)\} \subseteq S$. The probability of this event for one run of the inner loop in Algorithm 2 with the assumption that $d_{max} >= 2$ is given by:

$$P[\mathcal{E}_i(V)] = \frac{1}{d_{max}}(1 - \frac{1}{d_{max}})^{d_i} \geq \frac{1}{d_{max}}(1 - \frac{1}{d_{max}})^{d_{max}} \geq \frac{1}{d_{max}}\frac{1}{4}. \tag{24}$$

The last inequality holds for $d_{max} >= 2$ because $(1 - \frac{1}{x})^x \geq 0.25, \quad \forall x \geq 2$. Based on Lemma 1, the event $\mathcal{E}_i(V)$ implies that the directed edge $(V, X_i)$ will be present in $Tr(G[S])$. The outer loop runs for $4\alpha d_{max}\log(n)$ iterations and elements of the set $S$ are independently sampled. Hence, the probability of failure for all runs of the loop is given by:

$$P[(\mathcal{E}_i(V))^c] \leq (1 - \frac{1}{4 d_{max}})^{4\alpha d_{max}\log(n)} \leq e^{-\alpha\log(n)} = \frac{1}{n^\alpha}. \tag{25}$$

For a graph with total number of variables $n$, the total number of such bad events will be $\binom{n}{2}$. Union bounding the probability of bad events for every pair of variables, we have:

$$P[Failure] \leq \binom{n}{2} * \frac{1}{n^\alpha} \leq \frac{1}{n^{\alpha-2}}. \tag{26}$$

For inference, we need all interventional data sets $\mathbf{S}_i \subseteq \mathbf{X}$ such that $Pa(X_i) \subseteq \mathbf{S}_i$ & $X_i \notin \mathbf{S}_i$ for every treatment $X_i \in \mathbf{X}$ (Theorem 2 ). Suppose $p_i$ is the number of parents of a treatment variable $X_i$. We know that $p_i \leq d_{max}$. Let us define the event $\mathcal{E}_i = [Pa(X_i) \subseteq \mathbf{S}_i$ & $X_i \notin \mathbf{S}_i]$. The probability of this event for one iteration of the inner loop is given by:

$$P[\mathcal{E}_i] = \frac{1}{d_{max}}(1 - \frac{1}{d_{max}})^{p_i} \geq \frac{1}{d_{max}}(1 - \frac{1}{d_{max}})^{d_{max}} \geq \frac{1}{d_{max}}\frac{1}{4}. \tag{27}$$

The last inequality holds for $d_{max} >= 2$. The outer loop runs for $4\alpha d_{max}\log(n)$ iterations and the elements of the set $S$ are independently sampled. The probability of failure for all runs of the loop is given by:

$$P[\mathcal{E}_i^c] \leq (1 - \frac{1}{4 d_{max}})^{4\alpha d_{max}\log(n)} \leq e^{-\alpha\log(n)} = \frac{1}{n^\alpha}. \tag{28}$$

---

[1] $\sigma$ is any total order that is consistent with the partial order implied by the DAG, i.e., $\sigma(X) < \sigma(Y)$ iff X is an ancestor of Y.

Table 2: Two SCMs where $(U_1, U_2, U_Y)_{\mathcal{M}_1} \sim \mathcal{N}(0, \Sigma_{\mathcal{M}_1})$ and $(U_1, U_2, U_Y)_{\mathcal{M}_2} \sim \mathcal{N}(0, \Sigma_{\mathcal{M}_2})$.

| $\mathcal{M}_1$ | $\mathcal{M}_2$ |
|---|---|
| $X_1 = U_1$ | $X_1 = U_1$ |
| $X_2 = 2X_1 + U_2$ | $X_2 = X_1 + U_2$ |
| $Y = X_1 + X_2 + U_Y$ | $Y = X_1 + X_2 + U_Y$ |

$$\Sigma_{\mathcal{M}_1} = \begin{bmatrix} 1 & \frac{1}{4} & 0 \\ \frac{1}{4} & 2 & 0 \\ 0 & 0 & 1 \end{bmatrix} \quad \Sigma_{\mathcal{M}_2} = \begin{bmatrix} 1 & \frac{5}{4} & 0 \\ \frac{5}{4} & \frac{7}{2} & 0 \\ 0 & 0 & 1 \end{bmatrix}$$

Table 3: Two SCMs where $(U_1, U_2, U_Y)_{\mathcal{M}_3} \sim \mathcal{N}(0, \Sigma_{\mathcal{M}_3})$ and $(U_1, U_2, U_Y)_{\mathcal{M}_4} \sim \mathcal{N}(0, \Sigma_{\mathcal{M}_4})$.

| $\mathcal{M}_3$ | $\mathcal{M}_4$ |
|---|---|
| $X_1 = U_1$ | $X_1 = U_1$ |
| $X_2 = 3X_1 + U_2$ | $X_2 = 2X_1 + U_2$ |
| $Y = 2X_1 + X_2 + U_Y$ | $Y = 2X_1 + X_2 + U_Y$ |

$$\Sigma_{\mathcal{M}_3} = \begin{bmatrix} 1 & \frac{1}{2} & 0 \\ \frac{1}{2} & 3 & 0 \\ 0 & 0 & 1 \end{bmatrix} \quad \Sigma_{\mathcal{M}_4} = \begin{bmatrix} 1 & \frac{3}{2} & 0 \\ \frac{3}{2} & 5 & 0 \\ 0 & 0 & 1 \end{bmatrix}$$

For graph with total number of variables $n$, we will have $n$ of these bad events. The overall probability of missing any one of required data sets for the inference scheme can be determined by union bounding the probability of all such bad events:

$$P\left[ \bigcup_{i=1}^{n} \mathcal{E}_i^c \right] \le nP[\mathcal{E}_i^c] = \frac{1}{n^{\alpha-1}} \tag{29}$$

This shows that the probability of having all interventional data sets for inference after the algorithm runs is at least $1 - \frac{1}{n^{\alpha-1}}$.

### A.5 Additional Examples Showing Non-Identifiability of AOIs from Intervention on all Treatments and Observational Distribution in ANMs

In this section, we present additional examples of additive noise models that exhibit agreement on both the joint observational and joint interventional distributions. However, these models differ in their interventional distributions of subsets of treatment variables. Consider Table 2, which presents two SCMs labeled as $\mathcal{M}_1$ and $\mathcal{M}_2$.

The joint interventional distribution $do(X_1 = x_1, X_2 = x_2)$ is the same for both $\mathcal{M}_1$ and $\mathcal{M}_2$, i.e. $\mathcal{N}(x_1 + x_2, 1)$. The joint observational distribution for both models is a multivariate Gaussian i.e. $(X_1, X_2, Y)_{\mathcal{M}_1} \sim \mathcal{N}(0, \Sigma)$ and $(X_1, X_2, Y)_{\mathcal{M}_2} \sim \mathcal{N}(0, \Sigma)$. This is formally stated in equation 30. It is worth noting that due to the different structural equations for $X_2$ in the models, the interventional distribution $do(X_1)$ is different for the two models.

$$(X_1, X_2, Y)_{\mathcal{M}_1, \mathcal{M}_2} \sim \mathcal{N}(0, \Sigma), \quad \Sigma = \begin{bmatrix} 1 & \frac{9}{4} & \frac{13}{4} \\ \frac{9}{4} & 7 & \frac{37}{4} \\ \frac{13}{4} & \frac{37}{4} & \frac{27}{2} \end{bmatrix} \tag{30}$$

Table 3 presents another example of SCMs that exhibit agreement on both the joint observational and interventional distributions but differ in the interventional distribution $do(X_1)$. The joint interventional distribution $do(X_1 = x_1, X_2 = x_2)$ is the same for both, i.e. $\mathcal{N}(2x_1 + x_2, 1)$. Similar to the previous example, the joint observational distribution for both models is a multivariate Gaussian:

$$(X_1, X_2, Y)_{\mathcal{M}_3, \mathcal{M}_4} \sim \mathcal{N}(0, \Sigma), \quad \Sigma = \begin{bmatrix} 1 & \frac{7}{2} & \frac{11}{2} \\ \frac{7}{2} & 15 & 22 \\ \frac{11}{2} & 22 & 34 \end{bmatrix} \tag{31}$$

These examples highlights the possibility of constructing multiple such examples of additive noise models (ANMs) with Gaussian noise, where they align on the joint observational and joint interventional distributions but diverge on certain interventional distributions.

