# OpenReview forum: "Identification of Average Outcome under Interventions in Confounded Additive Noise Models"
_TMLR — Accepted by TMLR_

### Review · Reviewer_PHE7 · 2025-07-09

**Summary Of Contributions:**

The authors study the identification of the expected outcome under a joint intervention in additive noise models with potential confounders. The authors provide the identification criteria under the assumption that the potential confounders are additive and follows a Gaussian distribution, which implies that the treatment or the outcome (or both) are linearly dependent on the unmeasured confounders. The authors additionally propose a randomized algorithm to identify the target parameter through a sequence of interventions where they aim to simultaneously learn the transitive closure of the causal graph.

**Audience:**

Yes

**Claims And Evidence:**

No

**Requested Changes:**

1. The first three sentences of the abstract is very misleading. As the majority of the study on additive noise models in causal discovery focuses in the observational setting, the authors needs to make clear that they are using interventional dataset in the first three sentences. Saying that the unobserved confoudner follows a multivariate Gaussian is not an accurate description of the assumption. You need to highlight that it is additive. Your model in fact implies that the unmeasured confoudners follow linear Gaussian, which is a key distinction from observational settings, as it is known that this will not lead to a unique identificable causal graphs when using observational data alone.

2. In the abstract, you mentioned that “these interventions are sufficient to recover the causal structure between the observed variables.” However, in your randomized algorithm section, it seems that you are discovery the transitive closure of the causal graph instead. Which is not an accurate description.

3. The logic of the introduction is flawed: toward the second half of the first paragraph, the authors start by saying that RCTs are not feasible in real-world that’s why observational studies are useful. The authors then focus again on the interventional case, which contradicts in the logic earlier in the paper. The authors need to rewrite their introduction to focus on the interventional case, and clearly articulate the problem setting from the first paragraph.

4. The second and third sentence of the second paragraph is inaccurate: for example, when assuming ANM is linear, while the true DGP follows a nonlinear distribution will also yield nonindependent noises in the observational settings. When making these claims, the authors need to clearly articulate the problem setting.

5. The related work needs to be streamlined to categorize them into observational and interventional settings, and a direction contrast between the identifibility criteria is needed in this section to further clarify the setting of this paper. The current logic of this section again mixes the observational and internvetional cases without clear logic flow.

6. The sentence “Our work is focused on more general ANMs without any additional restrictions on the structural equations” is not true. The authors make strong assumptions on the structure equation of the unmeasured confounders, which is the focus of this paper.

7. It is useful for the authors to discuss in section 3 expicitly what their DGP on the unmeasured confounders assumes. For example, if the parents follow a nonlinear generating distribution, then this would entail that the parents of X_i must be all observed in this model. The authors currently present the structural equations as a less restrictive one. However, it makes strong assumptions of the allowable DGPs.

8. I found Theorem 1 poorly worded. “Is identifiable” in the sentence could be describing both W\in X and E[Y|do(X)]. Additionally, the conditions for identifiability is quite restrictive in the presense of the limited access to interventional data in the introduction, as it requires specific interventions on particular varaibles, which might not be feasible in practice.

9. In section 4.2, the authors start the paragraph with a “deterministic approach”. Now, for the randomized approach, are the authors trying to identify one potential outcome (one W) (as in constrast with the prior section which requires the identification under all W)? This point needs to be made clear as in the hypothesis testing literature, the distinction between the deterministic approach and randomized approach lies in the fact that the tests are now allowed to be performed sequentially to better identify the unknown true hypothesis. It is unclear whether learning the transitive closer of the graph factors in the improved complexity of the algorithm.

10. The transitive closer of the causal graph is known to be not as useful in inference. Additionally, it is unclear whether the graph is even well-specified under the presence of unmeasured confounding. The authors need to clarify this point.

**Strengths And Weaknesses:**

While the proposed problem is potentially interesting, I found the writing of the paper of low quality and the problem setting restrictive, which is not clearly articulated in the abstract or the introduction. The paper also contain inaccurate claims. The randomized algorithm is not clearly explained, hence i could not verify the validity of the approach under the presence of unmeasured confounders.

---

> ### Author Response · Authors · 2025-08-22
> **Rebuttal**
>
> We would like to thank the reviewer for their detailed review and comments. Below we address the reviewer’s concerns:
>
> **The first three sentences of the abstract are very misleading...**
>
> We clearly mention that our setting is additive noise models (ANMs), which implies that the noise is additive. We then state that the noise distribution is a multivariate Gaussian distribution. Together, this accurately describes our problem setup. We also explicitly state in the abstract: “We introduce a novel approach for estimating the average outcome under interventions (AOIs) for interventions on any subset of treatment variables and demonstrate that a small set of interventional distributions is sufficient to estimate all of them.” Thus, we clearly explain that our approach relies on interventional distributions for identification. In our opinion, the abstract clearly describes both the problem setup and our contributions.
>
> **In your randomized algorithm section, it seems that you are discovery the transitive closure of the causal graph instead.**
>
>
> Note that we propose two algorithms. Algorithm 1 learns the transitive closure of the causal graph under any arbitrary intervention, and Algorithm 2 randomly samples the intervention target sets $\mathbf{S}$ and accumulates the edges learned from the transitive closure of the post-interventional graphs $\mathcal{G}^{tc}_{\overline{\mathbf{S}}}$ to finally recover the observable graph structure.  In short, Algorithm 2 randomly selects $\mathbf{S}$, computes the transitive reduction of the post-interventional graphs, and accumulates all edges found in the transitive reduction across iterations. We theoretically prove in Supplementary Material Section A.3, Theorem 3, that Algorithm 2 is sufficient to recover the causal structure between the observed variables with high probability.
>
> **The logic of the introduction is flawed...**
>
> We appreciate the reviewer’s feedback regarding the logical flow of the introduction. Our intent was not to present observational and interventional data as contradictory, but rather to highlight the limitations of observational approaches and motivate why interventional data becomes essential in confounded settings. We agree that the introduction can be streamlined to emphasize the interventional case more clearly from the outset. We will revise the narrative accordingly to better align with the problem setting of our work
>
> **The second and third sentence of the second paragraph is inaccurate**
>
> Thank you for pointing this out. You are correct that non-independence of noise terms can also arise from model misspecification (e.g., assuming linear ANMs when the true DGP is nonlinear), not solely from hidden confounding. Our original statement was intended in the context of correctly specified additive noise models, where statistical independence of the noises is indeed equivalent to the absence of hidden confounders. To avoid confusion, we will revise the text to clarify that: independence of noise terms is a sufficient condition for no confounding under the assumption of correct model specification, whereas dependence among noise terms may reflect either hidden confounding or model misspecification. However, in this work we do not address the issue of model misspecification.
>
> **The related work needs to be streamlined to categorize them into observational and interventional settings, and a direction contrast between the identifibility criteria is needed in this section to further clarify the setting of this paper.**
>
>
> The related work section follows a logical progression. It begins by motivating the importance of causal inference across disciplines, then introduces the challenge of estimating causal effects from observational data and from combined observational/interventional data. Key results for observational settings are presented first, focusing on additive noise models (ANMs) and their identifiability under mild conditions (Peters et al., 2014; Rolland et al., 2022), including extensions to nonlinear mappings. The section then transitions to interventional settings, highlighting methods for combining observational and experimental data (Bareinboim and Pearl, 2012; Saengkyongam and Silva, 2020; Jeunen et al., 2022) and approaches that jointly leverage discovery and inference (Zhang et al., 2020). Importantly, in the presence of confounders, interventions are often required to identify causal effects, which naturally motivates this line of work. Finally, our work is positioned within this progression by considering more general ANMs without additional structural restrictions, where treatments may influence one another and Gaussian noise may have non-zero mean.
>
>
> To further improve clarity, in the revision we will explicitly categorize the prior work into observational versus interventional settings and provide a direct contrast of identifiability criteria, as suggested, which will make the distinctions and our contributions more transparent.

---

> ### Author Response · Authors · 2025-08-22
> **Rebuttal Continued...**
>
> **The sentence “Our work is focused on more general ANMs without any additional restrictions on the structural equations” is not true. The authors make strong assumptions on the structure equation of the unmeasured confounders, which is the focus of this paper.**
>
> We thank the reviewer for pointing this out. To clarify, in an ANM, the noise term is additive by definition, which is a structural assumption of the model. Our contribution lies in allowing treatments to have causal effects on one another, in contrast to prior work such as Jeunen et al. (2022), which assumes no causal interactions between treatments. While we do assume an additive structure for unmeasured confounders, this is inherent to the ANM framework rather than an additional restriction specific to our work. We will revise the sentence in the manuscript to avoid this confusion.
>
> **It is useful for the authors to discuss in section 3 expicitly what their DGP on the unmeasured confounders assumes. For example, if the parents follow a nonlinear generating distribution, then this would entail that the parents of $X_i$ must be all observed in this model. The authors currently present the structural equations as a less restrictive one. However, it makes strong assumptions of the allowable DGPs.**
>
> We already describe our problem setup in Section 3, more specifically in Equation (2). The assumption on unmeasured confounders is that they are sampled from a joint Gaussian distribution; that is, the latent variables, jointly represented as $\mathbf{U}$, follow a multivariate Gaussian distribution. This is also clearly stated in the abstract, where we mention that our problem setup is ANMs with multivariate Gaussian noise as the latent confounders.
>
> **Theorem 1 poorly worded  and Additionally, the conditions for identifiability is quite restrictive in the presence of the limited access to interventional data in the introduction, as it requires specific interventions on particular variables, which might not be feasible in practice.**
>
> We thank the reviewer for pointing this out. We will correct the wording in Theorem 1 and Theorem 2 by replacing “is identifiable” with “are identifiable.” Regarding the reviewer’s concern on restrictiveness, we clarify that in Theorem 1 we establish that interventions on parents are sufficient for identifying all AOIs. While this condition may appear strong, Theorem 2 extends our results by providing necessary and sufficient conditions, demonstrating that when one or more of the required interventional distributions are missing, only a subset of AOIs remain identifiable. Thus, our results precisely characterize the boundary between full and partial identifiability under limited interventional access.
>
> **Concerns regarding Section 4.2 and if we are just learning transitive closure.**
>
> We thank the reviewer for these comments. To clarify, our randomized approach in Section 4.2 is not limited to learning the transitive closure; rather, it aims to recover the full observable graph structure while reducing the number of required interventions compared to the deterministic approach. Algorithm 1 learns the transitive closure of the causal graph under any arbitrary intervention. Algorithm 2 randomly samples the intervention target sets $\mathbf{S}$ and accumulates the edges learned from the transitive closure of the post-interventional graphs $\mathcal{G}^{tc}_{\overline{\mathbf{S}}}$ to recover the observable graph structure. In short, Algorithm 2 repeatedly selects $\mathbf{S}$, computes the transitive reduction of the post-interventional graphs, and accumulates all edges found in the transitive reduction across iterations. We theoretically prove in Supplementary Material Section A.3 (Theorem 3) that Algorithm 2 is sufficient to recover the causal structure among the observed variables with high probability. The randomized sampling reduces the number of required interventions to learn the observable graph and infer all AOIs to polylogarithmic in the number of nodes, which is more efficient than the deterministic approach that requires a linear number of interventions in the worst case.
>
> We hope our response has clarified the reviewer’s concerns. We would be more than pleased to engage in further discussion if the reviewer has any additional concerns or questions.

---

> > ### Comment · Reviewer_PHE7 · 2025-08-27
> > **Updated Manuscript Request**
> >
> > I thank the authors for the clarification. I believe it would be beneficial to clarify the first three sentences of the abstract to clarify the interventional setting. Not everyone reads the paper work in this setting, and this can confuse readers who have only worked in observational settings. Although the authors addressed most of the concerns in the rebuttal, I would like to request an updated manuscript, with changes highlighted in a different color.

---

### Review · Reviewer_Nve7 · 2025-07-18

**Summary Of Contributions:**

The paper introduces a method to estimate the Average Outcome under Interventions (AOIs) in confounded Additive Noise Models (ANMs) where unobserved confounders follow a multivariate Gaussian distribution, requiring only a poly-logarithmic number of interventions to identify all causal effects even when the causal graph is unknown. This study addresses causal inference in confounded ANMs by developing a new algorithm to estimate AOIs for any subset of treatment variables, assuming multivariate Gaussian noise. The paper establishes necessary and sufficient conditions for the identifiability of AOIs, showing that the combination of observational data and targeted interventional distributions is enough to recover all AOIs. Specifically, they prove that interventions on each treatment’s parent set, along with a joint intervention on all treatments, enable full identification of causal effects. To reduce the intervention burden, they propose a randomized algorithm that selects intervention targets using a strongly separating set system and transitive closure of the causal graph. This approach learns both the observable graph structure and identifies sufficient interventional datasets for inference, requiring only $O(d_{max} \log^3n)$ interventions, where $d_{max}$ is the graph’s maximum degree and $n$ is the number of treatment variables. Empirical evaluations on synthetic data and a semi-synthetic healthcare Bayesian network validate the approach, showing accurate causal graph recovery and AOI estimation despite sample size limitations or deviations from Gaussian assumptions. Detailed proofs based on do-calculus establish the theoretical soundness of their identifiability conditions and algorithmic efficiency.

**Audience:**

Yes

**Claims And Evidence:**

Yes

**Requested Changes:**

Can the method be evaluated on additional datasets from domains such as healthcare, economics, or genomics to assess its practical utility and limitations?

Could you indicate how can the framework be extended or adapted to handle non-Gaussian noise distributions while retaining identifiability guarantees?

What algorithmic optimizations or parallelization strategies can improve scalability to large, high-dimensional, or dense causal graphs?

**Strengths And Weaknesses:**

The paper presents interesting results, rigorously establishing necessary and sufficient conditions for the identifiability of AOIs in confounded ANMs, supported by comprehensive mathematical proofs that leverage do-calculus and Gaussian noise properties. Furthermore, if the proposed randomized algorithm, which requires only a poly-logarithmic number of interventions, withstands tests, the paper significantly reduces the practical cost of experimentation. The tests on synthetic and semi-synthetic data based on the HEALTHCARE Bayesian network are non-trivial as well.

The main methodological limitation I find is the assumptions for the theoretical analysis. The identifiability proofs and methods rely on multivariate Gaussian noise assumptions, which limit their applicability to cases where noise deviates significantly from Gaussianity, a common occurrence in real-world data. A minor drawback is the size of the data used. The paper discusses scalability theoretically; empirical tests are limited to relatively small graphs (up to 20 treatment variables). The method’s performance on high-dimensional or densely connected graphs could strengthen the arguments in the paper.

---

> ### Author Response · Authors · 2025-08-19
> **Rebuttal**
>
> We would like to thank the reviewer for their detailed review and comments. Below we address the reviewer’s concerns:
>
> **Can the method be evaluated on additional datasets from domains such as healthcare, economics, or genomics to assess its
> practical utility and limitations?**
>
> We appreciate the reviewer’s suggestion regarding the evaluation of our method on additional real-world datasets from domains such as healthcare, economics, or genomics. Our primary contributions in this work are theoretical: we establish novel identifiability results for average outcomes under interventions (AOIs) in confounded additive noise models (ANMs) with gaussian noise, and we provide randomized algorithm that achieve poly-logarithmic intervention complexity. The experiments included in the paper are primarily designed to empirically validate these theoretical results in controlled settings. That said, the proposed inference scheme is applicable across domains, provided that the underlying modeling assumptions are satisfied—namely, the confounded ANM setting with multivariate Gaussian noise as outlined in our framework. If these assumptions are violated, the guarantees established in this work may no longer strictly apply. Nonetheless, the method can still serve as a heuristic in such cases. We also note that we already included an evaluation on a semi-synthetic dataset based on a healthcare Bayesian network to demonstrate practical applicability beyond purely synthetic graphs, including settings where our assumptions do not strictly hold.
>
>
> **Could you indicate how can the framework be extended or adapted to handle non-Gaussian noise distributions while retaining identifiability guarantees?**
>
> Our theoretical results require the latent confounder to follow multivariate Gaussian noise. We believe that the methodology and proof techniques can be
> extended to distributions where conditional expectations are completely characterized by the mean and variance, for instance, certain exponential family distributions. In the case of heavy-tailed or non-Gaussian distributions, it may
> be possible to bound the causal effects; however, since we focus on full identification results, we impose stricter assumptions for our theoretical guarantees to
> hold.
>
> **What algorithmic optimizations or parallelization strategies can improve scalability to large, high-dimensional, or dense causal graphs?**
>
> One parallelization strategy to improve the runtime for larger or denser causal graphs is to run the iterations in Algorithm 2—which repeatedly calls Algorithm 1 to learn the transitive closure under an arbitrarily sampled set—in parallel. Specifically, these calls to Algorithm 1 can be executed on different threads, and their results can then be aggregated.
>
> We hope our response has clarified the reviewer’s concerns. We would be more than pleased to engage in further discussion if the reviewer has any additional concerns or questions.

---

### Review · Reviewer_AFDw · 2025-08-17

**Summary Of Contributions:**

The authors studied a setting with multiple treatment variables, a single outcome, and potentially latent confounders. They assumed that the underlying causal model follows an additive noise model (ANM), where the latent confounders are jointly Gaussian, and it is allowed for the possibility of causal dependencies among the treatment variables. The main goal is to identify the causal effect of any subset of treatments on the outcome, referred to as the Average Outcome under Intervention (AOI), by carefully designing a suitable set of interventions. In Theorem 2, they established necessary and sufficient conditions on the interventional distributions required to identify all AOIs. Building on this, Theorem 3 demonstrates that a randomized procedure (Algorithm 2) can achieve this identification with high probability using at most $O(d_{\max} \log^2 n)$ interventions, where $d_{\max}$  is the maximum degree in the causal graph and $n$ is the number of treatment variables. Finally, the authors validated their theoretical contributions through experiments on both synthetic and semi-synthetic data.

**Audience:**

Yes

**Broader Impact Concerns:**

Nothing specific to be mentioned.

**Claims And Evidence:**

Yes

**Requested Changes:**

- How robust are the results if the latent confounders do not follow a multivariate Gaussian distribution—for example, in the presence of heavy-tailed or other non-Gaussian noise?

- To what extent does the additive noise assumption limit the applicability of the proposed framework in real-world systems where functional relationships may be more complex or non-additive?

- Although the number of experiments is poly-logarithmic, is the size of each intervention set within an experiment bounded, and what implications does this have for scalability in practice?

- Why is it practically important to identify the causal effect of any subset of treatments on the outcome, and in which concrete application domains would this be most relevant?

- In the last paragraph of page 3, the definition of the average treatment effect (ATE) is given over two possibly different sets $W_1$
 and $W_2$. Could the authors elaborate further on this definition and clarify its interpretation?

- In Section 4, the phrase “joint interventional distributions” seems misleading, as it appears the authors are actually referring to a set of interventional distributions. Could this terminology be clarified?

- In the last sentence of page 4, the authors write: “We show that with additional interventional datasets, including interventions on the parent sets of the treatment variables, we can make all possible AOIs in confounded ANMs identifiable.”
It seems that the intended meaning is interventions only on observed variables, since parent sets of treatment variables may include latent confounders, which are not directly intervenable. Could the authors confirm and rephrase this statement?

- When the notation $X_{int}$ is first introduced, please provide a clear definition.

- In the proof of Theorem 1 (Appendix A.2), please specify how the term $\Sigma_{yx} \Sigma_{xx}^{-1} V$ is derived.

- On page 14 of the proof, it is noted: “However, since we estimate the noise covariance matrix from data, if it is not invertible, we can apply Gaussian elimination to identify the dependent noise component(s).” Given that the proof is framed as an infinite-sample argument with access to the full distribution, why is it necessary to discuss finite-sample issues such as non-invertibility of the estimated covariance matrix?

**Strengths And Weaknesses:**

**Strengths**

- More General Setting: Unlike prior work that often assumes no causal relations among treatments or no latent confounding, this paper accommodates treatment–treatment dependencies and more general Gaussian confounding.
- Identifiability Results: Theorem 2 establishes rigorous sufficient and necessary conditions for identifying all average outcomes under interventions (AOIs).
- Algorithmic Contribution: The randomized algorithm (Theorem 3 and Algorithm 2) reduces the required number of interventions from linear to poly-logarithmic in the number of treatments.

**Weaknesses**
- Gaussian Assumption: The framework depends heavily on latent confounders following a multivariate Gaussian distribution, leaving its performance under non-Gaussian or heavy-tailed noise unexplored.
- Additive Noise Restriction: The assumption of additive noise throughout may limit applicability to real-world systems with more complex functional dependencies.
- Scalability in Practice: Although the number of experiments is poly-logarithmic, the size of each intervention set within an experiment is not clearly bounded.
- Limited Empirical Scope: The empirical evaluation focuses on small synthetic graphs and a semi-synthetic healthcare network.
- Motivation and Writing: The paper could more clearly articulate why identifying the causal effect of any subset of treatments on the outcome is practically important and in which applications. Additionally, some parts are difficult to follow.

---

> ### Author Response · Authors · 2025-08-18
> **Rebuttal**
>
> We would like to thank the reviewer for their detailed review and comments. Below we address the reviewer’s concerns:
>
> **Latent confounders do not follow a multivariate Gaussian distribution—for example, in the presence of heavy-tailed or other non-Gaussian noise**
>
> Our theoretical results require the latent confounder to follow multivariate Gaussian noise. We believe that the methodology and proof techniques can be extended to distributions where conditional expectations are completely characterized by the mean and variance, for instance, certain exponential family distributions. In the case of heavy-tailed or non-Gaussian distributions, it may be possible to bound the causal effects; however, since we focus on full identification results, we impose stricter assumptions for our theoretical guarantees to hold. In addition, in Section 5.2 we apply our inference scheme to cases where our assumptions do not hold, which results in higher estimation errors, but still provides reasonably precise estimates.
>
> **Additive noise assumption limit the applicability of the proposed framework in real-world systems**
>
> The additive noise assumption is commonly used in prior literature, and we discussed previous work on causal inference in additive noise models in detail in our paper. Note that the functional relationships between the observed nodes can be arbitrary — we place no restrictions on them. The only restriction is that the noise follows a Gaussian distribution and is additive. Our theoretical results will not hold when the noise is non-additive, and in such cases, additional interventional data may be required for full inference compared to our Theorem 2. In our manuscript, we explain with a simple example that while nonparametric identification will fail, our inference scheme is still able to identify the average causal effect under our parametric assumption. For more details, please refer to the second-to-last passage on page 5 of our paper.
>
> **Number of experiments is poly-logarithmic and size of each intervention set within an experiment**
>
>
>  We do not impose a bound on the maximum number of nodes within an intervention set, which enables the total number of experiments to be poly-logarithmic. In cases where one needs to limit the number of nodes within an intervention, one can use the $(n,k)$-separating system from Theorem 1 [1]. However, it may not be possible to ensure that the total number of experiments remains poly-logarithmic when imposing a limit on the size of intervention sets.
>
> [1] Shanmugam, Karthikeyan; Kocaoglu, Murat; Dimakis, Alexandros G.; and Vishwanath, Sriram. (2015). Learning Causal Graphs with Small Interventions. In Advances in Neural Information Processing Systems, 28, 2015.
>
> **Why is it practically important to identify the causal effect of any subset of treatments on the outcome**
>
> Identifying the causal effect of any subset of treatments is crucial because real-world interventions often occur in combinations, and their joint effects may differ from individual effects. This is particularly relevant in medicine, public policy, education, agriculture, and marketing, where multiple interventions are applied simultaneously, and the goal is to identify the best or optimal interventions through a series of experiments. The ability to determine effects for all possible intervention combinations, i.e., all possible subsets of treatments, using a much smaller number of interventional data sets, can be highly useful in such scenarios. Estimating these effects enables better decision-making, resource allocation, and the design of effective intervention strategies. We will add this motivation and practical importance of our work in the introduction of our revised manuscript.
>
> **In the last paragraph of page 3, the definition of the average treatment effect (ATE) is given over two possibly different sets**
>
> In our work, we define the average outcome under intervention (AOI) as $E[Y \mid do(\mathbf{W}=\mathbf{w})],
> $
> where $\mathbf{W}$ is any subset of the treatment variables, i.e., $\mathbf{W} \subseteq \mathbf{X}$.
> In previous literature, the average treatment effect (ATE), also sometimes referred to as the average causal effect (ACE), is defined as the difference:
> $E[Y \mid do(\mathbf{W}_1=\mathbf{w}_1)] - E[Y \mid do(\mathbf{W}_2 = \mathbf{w}_2)],
> $where $\mathbf{W}_1, \mathbf{W}_2 \subseteq \mathbf{X}$. Here, we are simply comparing our definition of AOI to another commonly studied causal quantity in the literature, the ATE, which in fact represents the difference of two AOIs. Thus, two different sets $\mathbf{W}_1$ and $\mathbf{W}_2$ are used in this context. Our proposed framework can also be used to identify the classic ATE or ACE, which is defined as the difference between two AOIs.

---

> ### Author Response · Authors · 2025-08-18
> **Rebuttal Continued...**
>
> **The phrase “joint interventional distributions” seems misleading**
>
> When we use the phrase joint interventional distribution, we actually refer to intervention on the entire treatment set $\mathbf{X}$, i.e., $do(\mathbf{X})$. We apologize that this phrase may be misleading, and we will replace it with ``intervention on the entire treatment set'' instead, to avoid any confusion.
>
> **Additional interventional datasets, including interventions on the parent sets of the treatment variables, we can make all possible AOIs in confounded ANMs identifiable.” It seems that the intended meaning is interventions only on observed variables**
>
> Throughout the paper, whenever we refer to the parents of any observed variable, we imply only the observed variables that are its parents in the actual graph. We clarify this in the first passage of Section 3 (Background), where we define the parents of a node.
>
> **Clarification on $\mathbf{X}_{int}$**
>
>
> The set $\mathbf{X}_{int} \subseteq \mathbf{X}$ is an arbitrary intervention set which is subset of treatment variables. There is no particular definition required here.  After this, in the proof, we discuss how one can identify the AOI for an arbitrary intervention set.
>
>
> **In the proof of Theorem 1 (Appendix A.2), please specify how the term $\Sigma_{yx}\Sigma_{xx}^{-1}\textbf{V}$ is evaluated**
>
>
> The matrices $\Sigma_{yx}$ and $\Sigma_{xx}$, formally defined in Equation (8), are composed of noise covariances. Equation (10) explains how the noise covariances can be identified using the available interventional data, while Equation (9) explains how the structural equations can be identified using the same data. Finally, Equation (11) illustrates how the vector $\mathbf{V}$ is evaluated.
>
> **Why is it necessary to discuss finite-sample issues such as non-invertibility of the estimated covariance matrix?**
>
>
> It is true that the proof is framed as an infinite-sample argument with access to the full distribution. In the case of multivariate Gaussian noise when at least one component is a linear combination of other components, the noise covariance matrix will be non-invertible even in the infinite-sample regime. Consequently, the matrix inverse computation required in Equation (8) may not be possible. Since we estimate the noise covariance matrix from data, if it is not invertible, we can apply Gaussian elimination to identify the dependent noise component(s), which can then be removed.
>
> We hope our response has clarified the reviewer’s concerns. We would be more than pleased to engage in further discussion if the reviewer has any additional concerns or questions.

---

> > ### Comment · Reviewer_AFDw · 2025-08-27
> >
> > Thank you for the response. I have a few follow-up suggestions:
> >
> > - It would be better to include the explanation of handling distributions beyond the Gaussian case in the main body.
> >
> > - In practice, to me, the number of individual interventions is important. Please clarify how many total individual interventions would be required under your design.
> >
> > - It would strengthen the paper to add motivation for why identifying the causal effect of all subsets of treatments on the outcome is valuable.
> >
> > - Regarding the non-invertibility of the estimated covariance matrix: the original explanation referred to finite-sample issues, which is not the case here. Please update the explanation accordingly, and also discuss the scenario where one component becomes a linear combination of the others.

---

> ### Author Response · Authors · 2025-08-27
> **Re.**
>
> Thank you for your comments. We will clarify the handling of distributions beyond the Gaussian case in the main text and provide motivation for why identifying the causal effect of all subsets of treatments on the outcome is important. Regarding the non-invertibility of the estimated covariance matrix, the original explanation referred to finite-sample issues, which do not apply here. We will update the discussion to emphasize the scenario where one noise component is a linear combination of others.
>
> Concerning the number of individual interventions, we do not impose a bound on the maximum number of nodes in an intervention set, allowing the total number of experiments to remain poly-logarithmic. If one wishes to restrict the size of each intervention set, a $(n,k)$ separating system from Theorem 1 from [1] can be used; however, the total number of experiments may no longer remain poly-logarithmic. If we count only single-node interventions, the total number of interventions scales as $\mathcal{O}(n)$. We will include this explanation in the paper.
>
> [1] Shanmugam, Karthikeyan; Kocaoglu, Murat; Dimakis, Alexandros G.; and Vishwanath, Sriram. (2015). Learning Causal Graphs with Small Interventions. In Advances in Neural Information Processing Systems, 28, 2015.

---

### Review · Reviewer_ScDV · 2025-08-19

**Summary Of Contributions:**

The paper studies confounded additive noise models (ANMs) with jointly non-isotropic Gaussian exogenous noise. This setting corresponds to a causal structure where treatment variables are confounded by hidden variables. The paper shows that all average outcomes under interventions (AOIs) are identifiable if and only if we have access to:
i) observational data,
ii) the joint intervention $P[Y|do(X)]$, and
iii) for each treatment $X_i$, an interventional distribution $P[V - S_i | do(S_i)]$ where $X_i \not \in S_i$ and $S_i$ contains all causal parents of $X_i$, i.e., $Pa(X_i) \subseteq S_i$.

Finally, the authors propose a randomized design which, given a bound on the maximum degree, requires only a polylogarithmic number of interventions to both learn the observable graph (via a direct application of Kocaoglu et al., 2017) and compute all AOIs. The paper assumes Gaussian noise and that no treatment is a descendant of the target variable $Y$.

**Audience:**

Yes

**Broader Impact Concerns:**

No comment.

**Claims And Evidence:**

Yes

**Requested Changes:**

- The identifiability condition in Theorem 2 is existential, yet in practice parents are unknown. The algorithm argues random $S$ will cover each parent set while excluding $X_i$ often enough, however the probability calculations with a transpernt mapping from these events to the returned $S$-collection needed by Theorem 2 is not discussed in the main text (only in appendix). Please make this link more concrete in the main body.
- Theorem 2 requires access to the full interventional distribution $P[Y \mid do(X)]$ in addition to the $S_i$ interventions. It is not clear whether the randomized routine that samples $S$’s can also recover the full intervention cases. Even though this would add only one additional deterministic intervention, it is worth mentioning.
- Starting from Definition 3, the paper becomes somewhat difficult to follow, both in the main body and in the appendix. I suggest that the authors include the preliminaries from Kocaoglu et al. (2017) and provide more detailed explanations.
- The paper alternates between the terms “transitive closure,” “transitive reduction,” and “observable graph.” With latent confounding, it is unclear what exactly is being learned? Moreover, the conditional independences used by Algorithm 1 implicitly require a weak faithfulness-like assumption after interventions, but this is not explicitly stated. The authors should clarify both the target graph and the assumptions in the post-interventional graph.
- Please include the details of the experiments, for example, the choice of estimators, as well as the source code for reproducibility purposes.

**Strengths And Weaknesses:**

Strengths:
- The AOI formalism and its identifiability are solid and extend the prior work of Jeunen et al. (2022) to settings where causal relationships between the treatment variables $X$ are allowed ($X \rightarrow X$ edges).
- By using strongly separating set systems, the paper provides a randomized algorithm that reduces the number of interventions to $O(d_{\max} \log^3 n)$ with high probability.
- Synthetic and semi-synthetic experiments show that the Structural Hamming Distance (SHD) and AOI error decrease as the number of samples or interventions increases, trending in the right direction.

Weakness:
- The main contributions, the identifiability results in Theorems 1 and 2, heavily rely on the Gaussianity of the noise. The key step in the identification procedure (page 14) depends on the linear properties of Gaussian conditioning, i.e., $\mathbb{E}[U_Y \mid X] = \mathbb{E}[U_Y] + \Sigma_{yx} \Sigma_{xx}^{-1} V$, and without the joint Gaussian assumption, the derivations break down.
- The non-identifiability example illustrated in the middle of page 5 does not rule out the non-Gaussian noise settings; instead, it seems that the non-identifiability stems from a violation of the additivity of the noise variable $U$.
- A major concern is the way “cost” is defined solely as the number of interventions, regardless of the size of the intervention set (number of knocked out variables in each intervention). Is this a valid criterion, and how can it be justified in practical applications? For example, if simultaneous interventions on all treatments are infeasible, can the results be adapted?

I defer my other concerns to the following section.

---

> ### Author Response · Authors · 2025-08-23
> **Rebuttal**
>
> We would like to thank the reviewer for their detailed review and comments. Below we address the reviewer’s concerns:
>
> **The key step in the identification procedure (page 14) depends on the linear properties of Gaussian conditioning**
>
> Our theoretical results require the latent confounder to follow multivariate Gaussian noise. We believe that the methodology and proof techniques can be
> extended to distributions where conditional expectations are completely characterized by the mean and variance, for instance, certain exponential family distributions. In the case of heavy-tailed or non-Gaussian distributions, it may
> be possible to bound the causal effects; however, since we focus on full identification results, we impose stricter assumptions for our theoretical guarantees to
> hold.
>
> **The “cost” is defined solely as the number of interventions, regardless of the size of the intervention set (number of knocked out variables in each intervention).**
>
> We do not impose a bound on the maximum number of nodes within an intervention set, which enables the total number of experiments to be poly-logarithmic. In cases where one needs to limit the number of nodes within an intervention, one can use the $(n,k)$-separating system from Theorem 1 [1]. However, it may not be possible to ensure that the total number of experiments remains poly-logarithmic when imposing a limit on the size of intervention sets.
>
> [1] Shanmugam, Karthikeyan; Kocaoglu, Murat; Dimakis, Alexandros G.; and Vishwanath, Sriram. (2015). Learning Causal Graphs with Small Interventions. In Advances in Neural Information Processing Systems, 28, 2015.
>
> **The identifiability condition in Theorem 2 is existential, yet in practice parents are unknown. The algorithm argues random sampling
>  will cover each parent set while excluding.**
>
> Although Theorem 2 may appear to suggest that, in the case of general confounded ANMs with $n$ treatments, $\mathcal{O}(n)$ interventional datasets are required to identify all possible AOIs, we show that this can in fact be reduced to polylogarithmic order by employing a randomized algorithm to select intervention targets. We also demonstrate that this set of interventions is sufficient for learning the observable causal graph, i.e., the induced graph between observed variables. Specifically, the proposed algorithm covers the parent sets through randomly sampled interventions, which enables us to reduce the number of required interventional datasets for both discovery and inference to polylogarithmic order in number of nodes ($n$). The detailed proof is included in the supplementary material (Section A.4). We will also include additional details, such as a proof sketch, in the main paper to highlight this more clearly.
>
> **Theorem 2 requires access to the full interventional distribution**
>
> Sorry for the omission. We will explicitly mention that we have access to the interventional distribution $P(Y \mid do(\mathbf{X}))$ in the body of Algorithm 2.

---

> ### Author Response · Authors · 2025-08-23
> **Rebuttal Continued...**
>
> **Starting from Definition 3, the paper becomes somewhat difficult to follow, both in the main body and in the appendix. With latent confounding, it is unclear what exactly is being learned? Moreover, the conditional independences used by Algorithm 1 implicitly require a weak faithfulness-like assumption after interventions, but this is not explicitly stated.**
>
> Our randomized approach in Section 4.2 is not limited to learning the transitive closure or the transitive reduction; rather, it aims to recover the full observable graph structure while reducing the number of required interventions compared to the deterministic approach. Algorithm 1 learns the transitive closure of the causal graph under any arbitrary intervention. Algorithm 2 randomly samples intervention target sets $\mathbf{S}$ and accumulates the edges learned from the transitive closure of the post-interventional graphs $\mathcal{G}^{\mathrm{tc}}_{\overline{\mathbf{S}}}$ to recover the observable graph structure. In short, Algorithm 2 repeatedly selects $\mathbf{S}$, computes the transitive reduction of the corresponding post-interventional graphs, and aggregates all edges found in the transitive reductions across iterations. We theoretically prove in Supplementary Material, Section A.3 (Theorem 3), that Algorithm 2 is sufficient to recover the causal structure among the observed variables with high probability. The randomized sampling reduces the number of required interventions to learn the observable graph and infer all AOIs to polylogarithmic in the number of nodes, which is more efficient than the deterministic approach that requires a linear number of interventions in the worst case.
>
> We define the terms transitive closure, transitive reduction, and observable graph clearly in our paper and use them to explain the approach. We will revise Section 4.2 to make it more explicit that the ultimate goal is to learn the observable graph, and that the tools used for learning it are the transitive closure and transitive reduction, thereby drawing a clear distinction between these objects. We also thank the reviewer for pointing out the need to state the (post-)interventional faithfulness assumption, which is commonly required by causal discovery algorithms. We apologize for omitting this assumption and we will also add the following clarification to the paper: We assume that the faithfulness assumption holds for both the observational and the post-interventional distributions. A probability distribution is said to be faithful to a graph if and only if every conditional independence statement in the distribution can be read off from the graph via $d$-separation.
>
>
> **Please include the details of the experiments, for example, the choice of estimators, as well as the source code for reproducibility purposes.**
>
> We have included the code in the Supplementary Material (zip file). In the camera-ready version, we will also provide a link to a public GitHub repository with the final code to ensure reproducibility. We describe the details regarding conditional independence testing in the Experiments section at the end of the first passage. In particular, the curse of dimensionality presents a challenge in testing conditional independence for continuous variables. To address this, we adopt a kernel matrix-based method that outperforms discretization-based approaches in terms of accuracy [2]. In addition to conditional independence testing, we also estimate the expected values required for computing the Average Outcomes under Interventions (AOIs). For this, we employ simple sample means depending on the experimental setting. We will add further clarifications in the revised Experiments section to make explicit the choice of estimators and their role in the reported results.
>
> [2] Kun Zhang, Jonas Peters, Dominik Janzing, and Bernhard Schölkopf. Kernel-based conditional independence
> test and application in causal discovery. arXiv preprint arXiv:1202.3775, 2012.
>
> We hope our response has clarified the reviewer’s concerns. We would be more than pleased to engage in further discussion if the reviewer has any additional concerns or questions.

---

### Author Response · Authors · 2025-08-27
**Revised Manuscript**

We have updated the manuscript, with changes highlighted in blue to indicate modifications based on the comments and suggestions of all the reviewers.

---

### Decision · Action_Editor_Qrpp · 2025-09-29

**Recommendation:** Accept with minor revision

**Additional Comments:**

For the camera-ready version, I encourage the authors to continue polishing readability and presentation, especially in technically dense sections (e.g., Section 4.2 and the proofs), to ensure accessibility for a broader causal inference audience.

**Audience:**

Yes

**Audience Explanation:**

AOI identifiability and efficient intervention design under latent confounding are central topics for the TMLR causal-inference readership. The characterization of when AOIs are identifiable in confounded ANMs and the randomized design with **poly-logarithmic intervention count** will interest researchers focused on theory, experimental design, and discovery under confounding. Despite scope limitations and presentation issues, the paper advances understanding of what is identifiable under explicit assumptions and provides algorithms with **provable guarantees**.

**Claims And Evidence:**

Yes

**Claims Explanation:**

This paper studies identifiability of Average Outcomes under Interventions (AOIs) in **confounded additive noise models (ANMs)** with multivariate Gaussian latents, allowing causal relations among treatments. It provides necessary and sufficient conditions for AOI identifiability and a randomized intervention design achieving **poly-logarithmic interventions** in the number of treatments, with proofs and synthetic/semi-synthetic validations.

Reviewers generally agree the theory is solid and extends prior work (e.g., Jeunen et al. 2022; Kocaoglu et al. 2017) in a meaningful way. That said, several caveats remain: (i) reliance on Gaussian additive confounding limits generality; (ii) writing and presentation, especially around Section 4.2 (closure vs reduction vs observable graph) and some terminology, need clearer exposition; (iii) empirical scope is limited and does not fully probe scalability; and (iv) parts of the technical novelty build on known separating-system ideas, which tempers originality. The rebuttal addressed many points and committed to clarifications, satisfying most (but not all) reviewer concerns.